

# Controls on surface soil drying rates observed by SMAP and simulated by the Noah land surface model

Peter J. Shellito[1], Eric E. Small[1]

[1]Geological Sciences, University of Colorado Boulder, Boulder, 80309, USA

*Correspondence to*: Peter J. Shellito (shellito@gmail.com)

**Abstract.** Drydown periods that follow precipitation events provide an opportunity to assess the mechanisms by which soil moisture dissipates from the land surface. We use SMAP (Soil Moisture Active Passive) observations and Noah simulations from drydown periods to quantify the role of soil moisture, potential evaporation, vegetation cover, and soil texture on soil drying rates. Rates are determined using finite differences over intervals of 1 to 3 days. In the Noah model, the drying rates

are a good approximation of direct soil evaporation rates. Data cover the domain of the North American Land Data Assimilation System phase 2 and span the first 1.8 years of SMAP's operation.

Drying of surface soil moisture observed by SMAP is faster than that simulated by Noah. SMAP drying is fastest when surface soil moisture levels are high, potential evaporation is high, and when vegetation cover is low. Soil texture plays a minor role in SMAP drying rates. Noah simulations show similar responses to soil moisture and potential evaporation, but

vegetation has a minimal effect and soil texture has a much larger effect compared to SMAP. When drying rates are normalized by potential evaporation, SMAP observations and Noah simulations both show that increases in vegetation cover lead to decreases in evaporative efficiency from the surface soil. However, the magnitude of this effect simulated by Noah is much weaker than that determined from SMAP observations.

## 1 Introduction

Though the volume of water is small, surface soil moisture generates outsized effects on the global water and energy balance (McColl et al., 2017b). Climate, weather, and flood predictions depend on soil moisture (Entekhabi et al., 1996; Viterbo and Betts, 1999). Feedback between the land and atmosphere can perpetuate soil moisture anomalies differently depending on the climatic regime (Koster et al., 2004; Tuttle and Salvucci, 2016). The duration of soil moisture anomalies depends on the drying rate of soil, which is controlled by complex interactions between soil hydrologic processes, atmospheric conditions,

and vegetation state (e.g., Rodriguez-Iturbe, 2000). Documenting the controls on soil drying is necessary to more fully understand the role of terrestrial hydrologic processes in the climate system.

Here, we use NASA's SMAP (Soil Moisture Active Passive) soil moisture observations and ancillary datasets to examine the relationship between soil drying and the factors expected to control direct evaporation from soil, including surface soil moisture, potential evaporation, soil texture, and vegetation cover. SMAP soil moisture is retrieved from L-band radiometer



measurements that are sensitive to water in the top 5 cm of the soil (Entekhabi et al., 2014). Thus, fluxes in and out of this surface soil layer control the drying rate determined from SMAP soil moisture time series (McColl et al., 2017a).

Precipitation wets the surface soil from above. Water that does not immediately runoff may dissipate via two paths: (1) returning to the atmosphere via evapotranspiration ($E_T$), or (2) moving deeper into the ground due to gravity and matric potential gradients. Excluding evaporation from wet canopy surfaces, $E_T$ is the sum of two fluxes: (1) direct evaporation from the soil surface, which typically occurs over a depth of several cm and (2) transpiration via plant stomata, which removes water from throughout the root zone (Campbell and Norman, 1998; Monteith and Unsworth, 2013).

To evaporate water, energy is required to overcome the latent heat of vaporization, and a vapor sink is required for the atmosphere to absorb it. Over open water or bare soil, these processes depend on surface vapor pressure, atmospheric vapor pressure, radiation, albedo, and wind velocity (Mahrt and Ek, 1984; Penman, 1948). These quantities contribute to the calculation of potential evaporation (PE) rate, or the atmospheric demand for moisture. Evaporation from vegetated areas must further be scaled by the stomatal and aerodynamic resistances associated with the vegetation itself (Wang and Dickinson, 2012).

$E_T$ also depends on the moisture supply (volumetric soil moisture, VSM) in the soil. In the Noah land surface model (LSM) and other models, the moisture stress on direct evaporation is represented as a piecewise function that depends on two VSM thresholds. Above field capacity, evaporation proceeds at its potential rate (stage one evaporation); below a residual soil moisture content, no evaporation occurs; between the two thresholds, evaporation depends on relative moisture content (stage two evaporation) (Allen, 2000; Chen and Dudhia, 2001). Using SMAP data, *McColl et al.* (2017a) showed stage 2 evaporation is dominant at continental scales.

Any vegetation present on the land surface introduces two counteracting effects on soil drying that do not exist for bare soil surfaces. Direct evaporation decreases because of shading from the canopy that intercepts solar radiation (Mahfouf and Noilhan, 1991). Transpiration draws moisture from the root zone into the atmosphere, with the depth of this flux depending on root distribution (Schenk and Jackson, 2002).

The soil texture of the land surface, and thus its hydraulic properties, also influences soil drying. Compared with fine textured soils (clay), coarser soils (sand) have more well-connected pores allowing water to leave the system more easily at a given moisture level (Campbell, 1974; Clapp and Hornberger, 1978; Cosby et al., 1984; Van Genuchten, 1980). These differences cause volumetric soil moisture to be lower for coarse textures than for fine (Laio et al., 2001). Despite the efforts that have gone into quantifying and modeling the effects of textural differences on soil moisture and heat fluxes, there remains much variability within each texture class, so assigning model parameter values based on continental scale soil maps has proven to be problematic (Gutmann and Small, 2005; Xia et al., 2015b). Nevertheless, the Noah LSM and other models use soil texture to assign soil hydraulic properties, which dictate the infiltration and redistribution of moisture in the soil column (Chen and Dudhia, 2001).

Direct evaporation from soil is notoriously difficult to measure. Lysimeters and chamber measurements provide information over extremely small areas (~10 $m^2$ or less) (e.g., Herbst et al., 1996; Stannard and Weltz, 2006). It is more challenging to



monitor at larger scales, but documenting evaporation and its contribution to total $E_T$ is necessary to more completely understand and model the flux of water from the land surface to the atmosphere (Kool et al., 2014).

Soil drying rates determined from satellite-based observations can provide an estimate of surface evaporation rates on a larger scale (McColl et al., 2017a). This requires that the depth supplying evaporation is sampled by the sensor and that

vertical redistribution within the soil is negligible. Evaporation largely draws from the top several cm of the soil column, within the sensing depth of L-band radiometers (Njoku and Entekhabi, 1996), although under extreme dry conditions the evaporative front will propagate down to greater depths. Drainage from the surface soil occurs on timescales of hours. Thus drainage can be ignored when considering drying intervals on a timescale of days, as is documented by the 1−3 day SMAP soil moisture observations (Chan et al., 2016). Assuming drainage is negligible outside of rainy intervals is a common

assumption in models of soil moisture dynamics (Federer et al., 2003; Guswa et al., 2002; e.g., Laio et al., 2001; McColl et al., 2017a; Porporato et al., 2004). We make this assumption here.

In situ observations have allowed for investigation of how different environmental factors control soil drying rates and thus evaporation rates (Cavanaugh et al., 2011; Detto et al., 2006; e.g., Kurc and Small, 2004, 2007). Only recently has satellite remote sensing of soil moisture advanced sufficiently to make it possible to monitor drying rates at large scales (Entekhabi et

al., 2010; Kerr, 2006), thus allowing scientists to evaluate the physical controls on soil drying across a wider range of conditions. *McColl et al.* (2017a) studied global soil drying dynamics by fitting SMAP surface soil moisture observations to an exponential decay function. They found shorter drying timescales in areas that have higher aridity indexes and higher soil sand content, although the effects of soil texture were relatively minor. Their results confirm the expected roles of atmospheric demand and soil texture on soil drying, and thus presumably also direct evaporation. They noted, however, that

there was substantial unexplained variance in drying timescales, citing vegetation as a likely factor. In addition, *McColl et al.* (2017a) only considered static descriptors of the physical environment at each location: neither aridity index nor soil texture vary through time. Yet, the soil drying process represents a dynamic interplay between hydrologic, climatic and ecosystem processes (Rodriguez-Iturbe, 2000).

Here, we use SMAP soil moisture observations and ancillary datasets to examine how the following factors control soil

drying: soil moisture, PE, soil texture, and vegetation cover. We do not fit an exponential decay model to drydown events as in *McColl et al.* (2017a). Instead we use calculations of changes in surface soil moisture through time, which *Shellito et al.* (2016b) showed to provide a similar depiction of soil drying as the exponential approach.

Our study builds upon *McColl et al.* (2017a) in two ways. First, we examine the effects of dynamic controls on soil drying, rather than static covariates. Soil moisture supply, PE rate, and vegetation cover are observed coincident with both the time

and location of the SMAP drying observations. This approach allows us to obtain results indicative of time-varying hydrological mechanisms and processes, rather than of the overall climate (e.g., aridity index) or soil type in each location.

Second, we compare our results to output from the Noah LSM (Ek et al., 2003; Xia et al., 2012a). Noah development and validation have been more focused on reproducing heat fluxes and runoff than soil moisture (Chen and Dudhia, 2001; Xia et al., 2012a, 2012b). A comparison between Noah and the North American Soil Moisture Database shows it is able to capture





broad features of soil moisture variations (Xia et al., 2015a). By analyzing Noah soil drying rates alongside the observations from SMAP, we can offer guidance into the strengths and limitations of Noah-simulated soil moisture dynamics. In addition, the model frames our understanding of the relationship between surface drying rates and surface evaporation rates.

## 2 Data and Methods

### 2.1 Data

Our study utilizes SMAP, NLDAS-2 (North American Land Data Assimilation System phase 2), and NDVI (normalized difference vegetation index) data from the nearly two-year period since SMAP began operation: 31 March 2015, through 27 January 2017.

Although SMAP and NDVI data are available globally, the one-eighth degree NLDAS-2 forcing and simulation data cover
only North America. Therefore, our study is limited to the continental land mass found between longitudes 124.9° and 67.1° W and latitudes 25.1° and 52.9° N (Figure 1).

### 2.1.1 SMAP retrievals

SMAP was launched in January 2015 and provides morning and evening (06:00 and 18:00 LT) estimates of VSM, in $cm^3$ $cm^{-3}$, between 0 and 5 cm globally every 1–3 days (Entekhabi et al., 2014). Retrievals are estimates of soil moisture based on
passive microwave (1.41 GHz) brightness temperature as described in *Entekhabi et al.* (2014). We use the "enhanced" level 3 soil moisture data product, Version 1, which is available from the National Snow and Ice Data Center (O'Neill et al., 2016). The SMAP radiometer has a native spatial resolution of 36 km, but this product utilizes the Backus-Gilbert optimal interpolation algorithm to post soil moisture retrievals onto the 9 km Equal-Area Scalable Earth grid version 2 (EASE-2) (O'Neill et al., 2016). The enhanced resolution version reveals spatial features not apparent in the 36 km standard product
and similarly meets the mission goal of 0.040 $cm^3$ $cm^{-3}$ unbiased root mean squared error [*Chan et al.*, submitted to *Remote Sensing of the Environment*]. We use only AM overpasses because the SMAP algorithm assigns one temperature to both the soil and its overlying canopy, a condition that is best met in the morning hours (Entekhabi et al., 2014; Jackson et al., 2012). We exclude data that have been flagged for uncertain quality due to dense vegetation (>5 kg $m^{-2}$), mountainous terrain (>3° slope standard deviation), and >5 % of the sensing area comprising frozen ground, snow, ice, precipitation, or static water.
These exclusions decrease the number of SMAP observations by 56.5 %, mostly because of vegetation in the eastern portion of North America. Figure 1a shows the number of SMAP observations used in this study, after removing flagged data. The domain consists of 136 422 SMAP pixels. Un-flagged SMAP observations are found in 59 % of the pixels, so these 79 987 "active" SMAP pixels are the focus of our study.




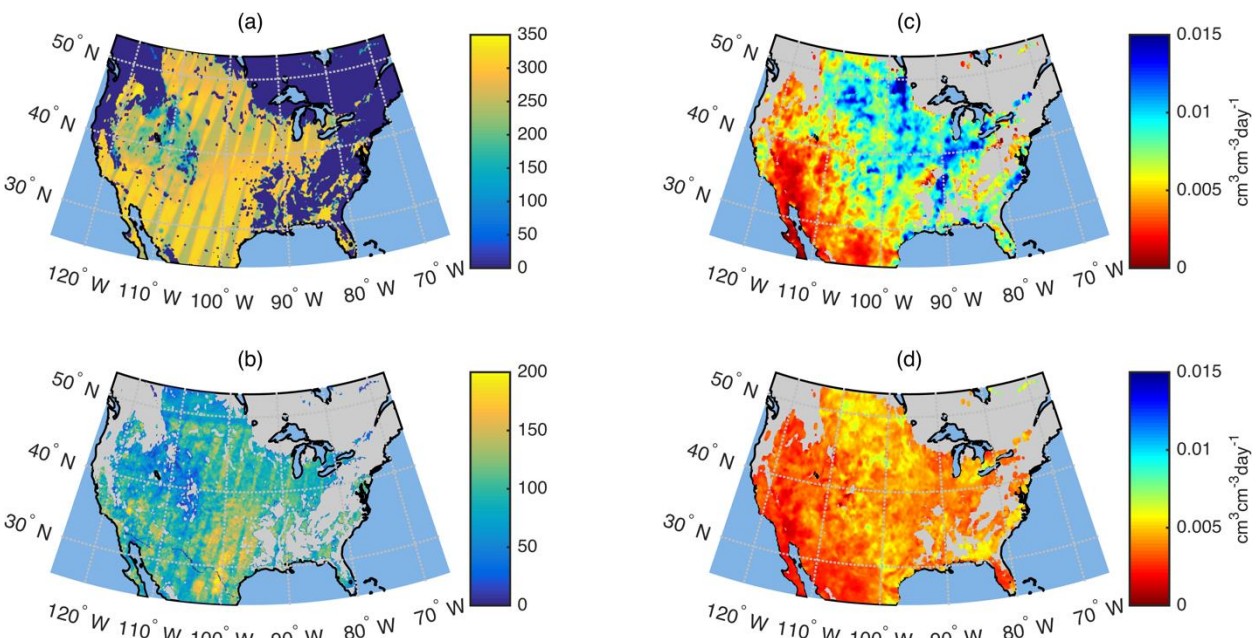

**Figure 1: (a) Number of SMAP observations used in this study. (b) Number of drying rates calculated from SMAP observations and Noah simulations. (c) Median of SMAP drying rates. (d) Median of Noah drying rates. Drying rates are expressed as changes in volume per day ($cm^3$ $cm^{-3}$ $day^{-1}$).**

### 2.1.2 NLDAS-2 precipitation, PE, and soil texture

We use precipitation and PE data from the NLDAS-2 primary forcing fields (Xia et al., 2012b). Precipitation is from the NCEP Climate Prediction Center's unified gauge-based precipitation, which has been adjusted for orographic effects (Daly et al., 1994). Other meteorological forcings are from the National Center for Environmental Prediction (NCEP) North American Regional Reanalysis (NARR), interpolated to the NLDAS-2 one-eighth degree grid and disaggregated to hourly frequency (Cosgrove et al., 2003). PE is calculated from those forcings using the modified Penman scheme of *Mahrt and Ek* (1984).

Within the United States, NLDAS-2 provides a gridded soil texture field derived from 1 km State Soil Geographic (STATSGO) data (Miller and White, 1998; Mitchell et al., 2004). Although there are 15 categories, some types (silt, sandy clay loam, sandy clay, silty clay, organic materials, water, and bedrock) individually occupy less than 3 % of the domain. We therefore focus on the four most common textures: loam (26.0 %), silt loam (25.9 %), sandy loam (23.0 %), and sand (6.8 %).

### 2.1.3 Vegetation data

NASA's Terra and Aqua satellites carry the MODIS instrument and provide NDVI data every 16 days globally, at a resolution of 1 km. NDVI is linearly interpolated in the days between retrievals and up-scaled to match the SMAP grid by taking the arithmetic mean of the MODIS cells contained in each SMAP pixel.



### 2.1.4 NLDAS-2 Noah simulations

As part of the NLDAS-2 project, Noah LSM simulations are run from 1979 to present and archived at the Goddard Earth Sciences Data and Information Services Center (GES DISC) (Xia et al., 2012b). Our study utilizes surface evaporation rates and surface soil moisture from these simulations. The top soil layer in the Noah LSM is 0–10 cm, twice the approximate

sensing depth of SMAP. Soil moisture values are converted from kg m$^{-2}$ to cm$^3$ cm$^{-3}$ to be consistent with SMAP units.

Noah partitions evapotranspiration between surface evaporation and transpiration through a parameterization of the fraction of land that has green vegetation on it ($F_G$) (Chen and Dudhia, 2001). This parameter is defined over a 1° grid by *Gutman and Ignatov* (1998) using 5 years of NDVI data according to Eq. (1)

$$F_G = \frac{NDVI - NDVI_0}{NDVI_\infty - NDVI_0},$$  (1)

where $NDVI_0$ is over bare soil and $NDVI_\infty$ is over dense vegetation. Thus, Noah simulations use vegetation climatology at each point, not vegetation observations. In contrast, our analyses are based on NDVI observations themselves (Sect. 2.1.3). To understand how this difference might affect the Noah results presented here, we additionally look at the $F_G$ values in our study domain, which are provided as part of the NLDAS-2 Noah simulation dataset.

### 2.1.5 Supplementary simulations

In addition to Noah simulations from NLDAS-2, we run the most recent Noah version (3.4.1) in two configurations for six US locations: Fort Cobb, OK, Little River, GA, Little Washita, OK, Marena, OK, St. Joesph's, IN, and Walnut Gulch, AZ. One simulation uses the default 0–10 cm layer 1 depth. The other uses a modified 0–5 cm layer 1 depth. In the latter case, layer two has been increased by 5 cm (to 5–40 cm) to keep total model soil depth unchanged. Further information regarding the location, soil type, and vegetation cover in these locations can be found in *Shellito et al.*, (2016a). The forcing data and

parameter values for these simulations are taken from the NLDAS-2 data corresponding to each location.

### 2.2 Methods

### 2.2.1 Pixel matchups

Our analyses require matching simulated or observed drying rates with concurrent observations of PE, vegetation, and soil type. These datasets come from different sources (NLDAS-2 and MODIS) and therefore have been re-gridded onto the

SMAP EASE-2 grid.

MODIS grid cells, which are finer than SMAP's grid, have been averaged together within each SMAP pixel. The NLDAS-2 grid is only slightly coarser than SMAP's grid, so occasionally the same data will be mapped into two SMAP pixels. Though this is not ideal, it is preferable to basing our analysis on the NLDAS-2 grid, which would force us to exclude some SMAP pixels or blend them with their neighbor when they fall within the same NLDAS-2 pixel.





### 2.2.2 Drydown periods

We utilize the precipitation field in the NLDAS-2 forcing dataset to select drydowns for our analysis. Following *Shellito et al.* (2016b), a drydown is defined by a dry period that follows a soil wetting event. We automate this selection process for all pixels according to the following logic: (1) the event precipitation depth must surpass 5 mm in a 24 hour period; (2) the dry period must begin after the event precipitation stops and end a day before 3 mm or more additional precipitation accumulates; and (3) the dry period must be at least 3 days long (Figure 2). We note this differs from *McColl et al.* (2017a) that based the identification of dry down periods on the soil moisture time series alone, and did not use any precipitation data.

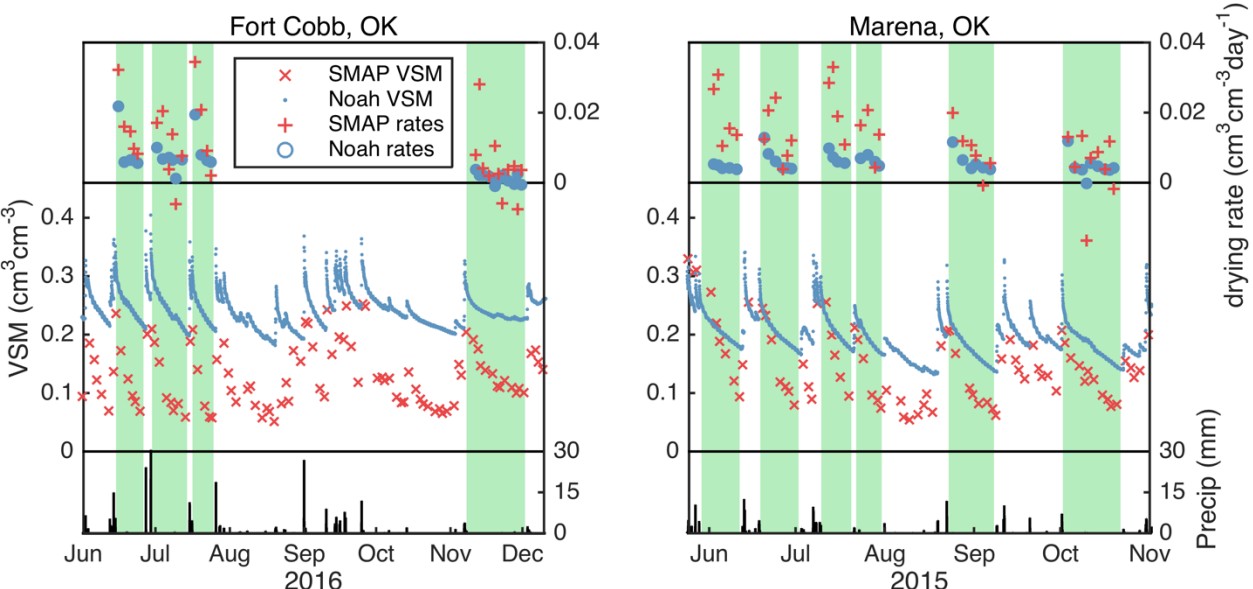

**Figure 2: Precipitation (bottom), soil moisture (middle), and drying rates (top) for two sites in Oklahoma: Fort Cobb (98.573º W, 35.342º E) and Marena (-97.217º W, 36.063º E). Drydowns are indicated with green shading.**

### 2.2.3 Calculation of drying rates

With both SMAP and Noah soil moisture data, we calculate soil drying rates that are contained within the drydown periods. As in *Shellito et al.*, (2016b) we use a simple finite differences approach:

$$\frac{d\theta}{dt} = \frac{\theta_{n+1} - \theta_n}{t_{n+1} - t_n}. \tag{2}$$

$\theta$ is surface soil moisture content (cm$^3$ cm$^{-3}$), $t$ is time (days), and n and n+$I$ correspond to consecutive observations (Figure 3). SMAP data are available every 1–3 days, so drying rates span at least 24 hours. Although simulated data are available hourly, we only use Noah soil moisture values that are concurrent with SMAP observations. This ensures that sampling frequency will not affect our comparison to Noah.





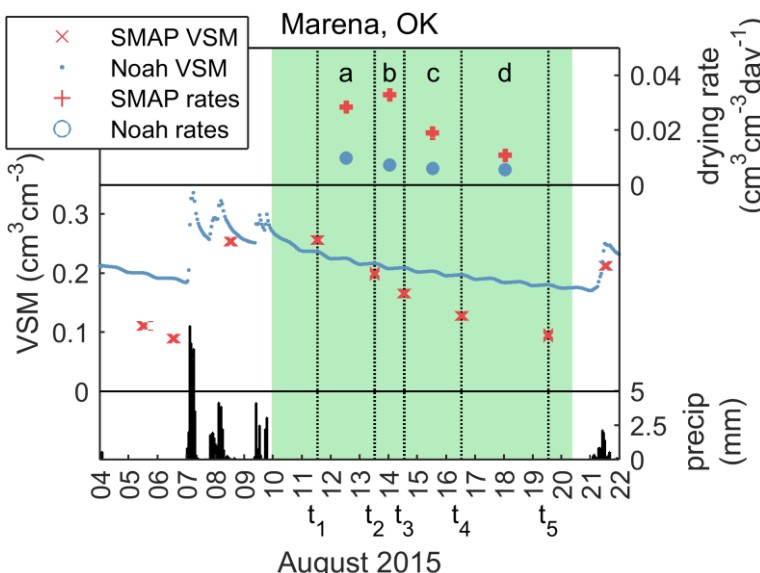

**Figure 3: Detail from one drydown period. Drying rates a, b, c, and d are the changes in soil moisture divided by the width of each calculation interval, which is the time between consecutive SMAP overpasses. These are shown with dotted lines ($t_1$ to $t_2$, $t_2$ to $t_3$, etc). Each drydown's PE and NDVI are taken as the average observed within the respective calculation intervals.**

Our analysis produces 4 738 702 drying rates for both SMAP and Noah, or an average of 75.2 per active SMAP pixel (Figure 1b). Figure 2 shows two representative soil moisture time series, the drydown periods contained therein, and the associated drying rates that have been calculated.

Preliminary analyses showed that SMAP observations can occasionally reach and stay at a maximum value, producing drying rates of exactly 0 mm day$^{-1}$. This is an artifact of the SMAP algorithm and does not reflect the drying process. Cases

where VSM stays constant at 0.5 during identified drydowns have been excluded.

The units of drying as calculated from SMAP surface soil moisture are cm$^3$ cm$^{-3}$ day$^{-1}$, a measure of the change in moisture volume through time. We express soil drying in two other ways. First, it is also useful to express drying rate in terms of the depth of water lost from the surface, to match the units of potential and actual evaporation. To covert from cm$^3$ cm$^{-3}$ day$^{-1}$ (volumetric change per day) to mm day$^{-1}$, we multiply SMAP-observed drying rates by its nominal sensing depth (50 mm

(Entekhabi et al., 2014)), and Noah-simulated drying rates by its layer thickness (100 mm). Second, we convert the drying rate to an evaporative efficiency: the fraction of PE that is realized by the above-calculated equivalent evaporation rate.

The surface soil drying rate is a proxy for direct evaporation from the soil. Most vertical redistribution of precipitation occurs within hours of rainfall, thus these intervals are typically not included in the SMAP overpass intervals used to calculate drying rates (Figure 3). In the Noah LSM, ET is partitioned between evaporation and transpiration, the former of which

removes moisture only from the top model layer. In Figure 4 we evaluate the correspondence of Noah evaporation with the drying rate calculated from the simulated soil moisture time series. Noah drying rates have a nearly 1:1 relationship with evaporation rates. We therefore call the drying rate in mm day$^{-1}$ the "equivalent evaporation rate" of the land surface. Points lying above the 1:1 line indicate instances when transpiration and/or drainage are contributing to soil drying in addition to





evaporation. The role of transpiration, however, is small because most plant types have 90 % of their roots below 10 cm (Ek et al., 2003). In addition, Figure 4 shows that the majority of points (58.4 %) lie below the 1:1 line, indicating that during most drydown periods, the magnitude of capillary rise is larger than drainage and transpiration combined. The equivalent evaporation rate is therefore a slightly conservative estimate (on average).

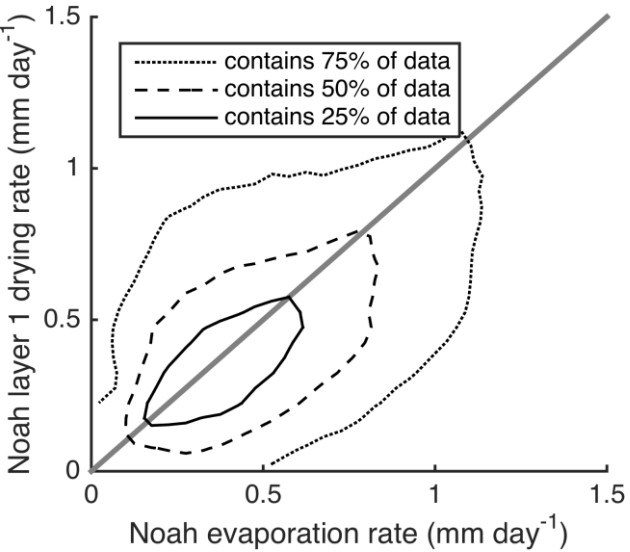

**Figure 4: Contours from a scatter plot showing correspondence of Noah layer 1 drying and evaporation rates (showing all 5+ million data points is prohibitive). 1:1 line is shown in gray. R=0.47.**

There is no equivalent way to test if SMAP drying rates also correspond to direct evaporation rates. However, it is reasonable to assume that the drying rates are similarly a proxy for direct evaporation. The equivalent evaporation rate of SMAP is based on a sensing depth that is half that of Noah's surface layer, so it will be a more conservative estimate than Noah's evaporation rate. There are fewer roots in this layer and thus less water loss from transpiration. In addition, any evaporation from below the 5 cm sensing depth is not accounted for. Because the evaporative efficiency is derived from equivalent evaporation rate, this quantity will also be conservative, particularly for SMAP.

### 2.2.4 Effects of meteorologic conditions and land surface states

We quantify the roles of time since rainfall, surface moisture content, PE, vegetation, and soil texture on drying dynamics using the data described in Sect. 2.1. For each rate calculated (Eq. 2), we record the arithmetic mean of each quantity between $t_n$ and $t_{n+1}$ (Figure 3).

With these data, we provide continent-wide summaries of the correlation between each variable and the different measures of soil drying. We present our results in terms of how median drying rates change with each variable. In all cases, bootstrapping is employed to estimate standard errors (Efron and Tibshirani, 1993). Bootstrapped statistics are generated using 500 instances of 100 random samples.





### 2.2.5 Effects of depth

SMAP and Noah do not represent identical soil depths. We use the supplementary simulations described in Sect. 2.1.5 to assess how Noah model soil moisture dynamics would change if its first layer depth were 0–5 cm, instead of 0–10 cm.

## 3. Results

5    The drying rates observed by SMAP and simulated by Noah decrease with time since the end of the previous precipitation event, consistent with prior results using similar data (McColl et al., 2017a; Shellito et al., 2016b). Thus, as the soil dries following precipitation (Figure 5a), the drying rate and equivalent evaporation rate also decrease. The form of the relationship between drying rate and VSM are described in more detail below.



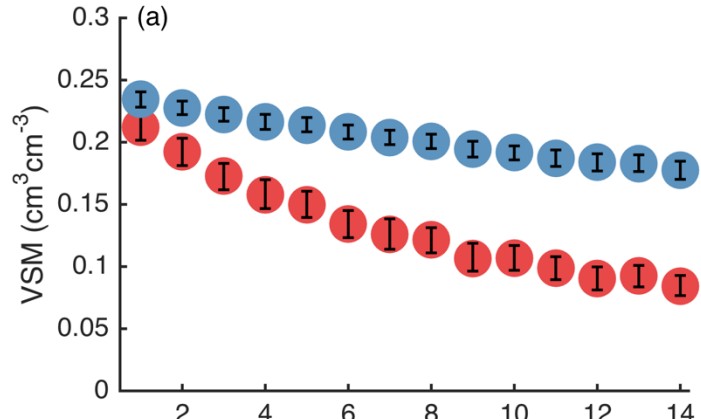

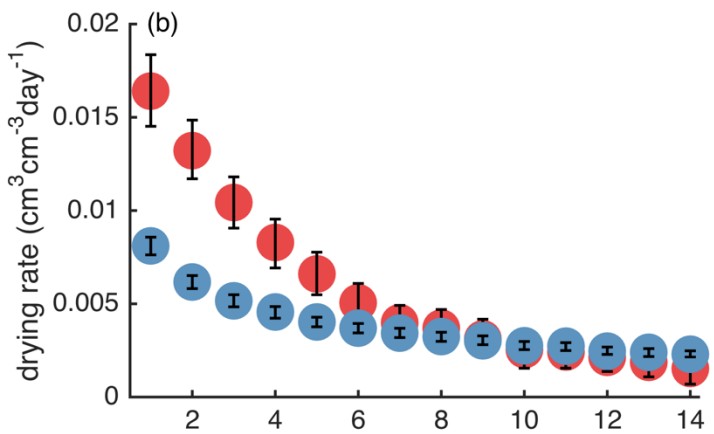

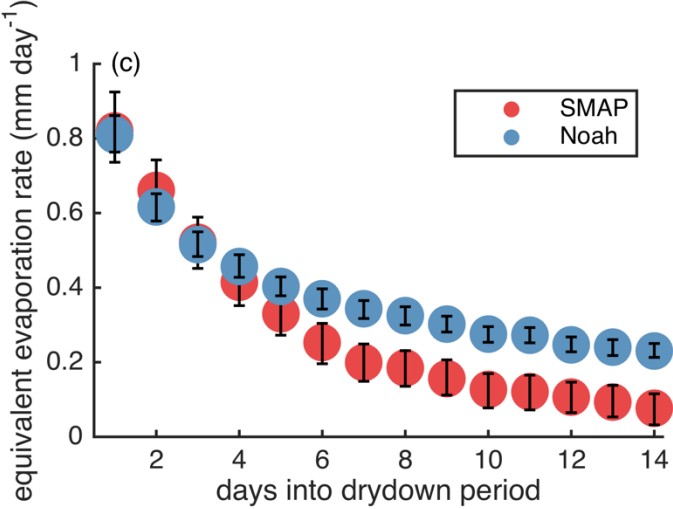



**Figure 5: Soil moisture (a), drying rate (b), and equivalent evaporation rate (c) as a function of days into the drydown period. Markers show median values. Error bars show standard error.**

Although SMAP and Noah both show that drying slows with increased time since rainfall, considerable differences exist. Directly following precipitation, drying rates are higher based on SMAP data than simulated by Noah, by a factor of two (Figure 5b). However, SMAP drying rates decrease more quickly through time. After about 10 days, Noah drying rates are equal to or slightly faster than SMAP rates. The SMAP and Noah equivalent evaporation rates are similar (~0.8 mm day$^{-1}$) directly following rainfall (Figure 5c). Evaporation from Noah occurs over twice the approximate SMAP sensing depth, making up for the differences in drying rate (Fig. 5b) and contributing to higher equivalent evaporation rates than from SMAP after ~5 days.

In addition to the more rapid decrease in drying rates observed with SMAP shown in Figure 5, there are also obvious geographic differences between SMAP and Noah drying rates (Figure 1c and d). Median values of drying rates are calculated in a 25 pixel (5 by 5) moving window. Data are displayed at the center of each window whenever at least 100 drying rates are contributing to the median. Median drying rates from SMAP and Noah are similar in the southwest region of the continent where precipitation events are small and infrequent. In these areas, soil moisture remains near its residual value for much of the time. As a result, the large differences that exist immediately after rain events (Figure 5b) occur infrequently and do not influence the median values shown in Figure 1c and d. In wetter regions, median drying rates from SMAP are considerably faster than simulated by Noah, at least partly because data from several days after precipitation events affects the median value.

To further investigate the causes of the different drying rates shown above, we consider the variables we understand to control the soil drying process: moisture supply (surface VSM), atmospheric demand (PE), vegetation cover, and soil texture.

### 3.1 Soil moisture and PE

We now investigate how surface soil moisture and PE influence drying rates (Figure 6). We have divided the drying rate data into terciles according to PE rate. A single PE value is used for each pair of corresponding SMAP and Noah drying rates, so the terciles are composed of an identical group (in terms of location and time) of observations from SMAP and Noah. We use 10 equal-width bins of increasing soil moisture. VSM is not equal for corresponding SMAP and Noah observations.





**Figure 6: Drying rates and efficiencies for SMAP (panels a-c) and Noah (panels d-f) as a function of surface soil moisture content and three PE terciles. Top row shows drying rates, middle row shows equivalent evaporation rates, and bottom row shows evaporative efficiencies. Markers show median values. Error bars show standard error.**



SMAP drying rates are highest when surface VSM and PE are high. The slowest rates are found when the soil is dry, regardless of PE. Drying rates monotonically increase with surface soil moisture, except in the low PE tercile, where there is a plateau in drying rates for soil moisture exceeding 0.15 cm³ cm⁻³. Across most of the range of soil moisture values, drying rates are clearly greater when PE is high.

Noah drying rates shows similar, though smaller, responses to PE and soil moisture. However, there is no plateau in Noah drying rate when PE is low and soil moisture is high. SMAP shows a larger sensitivity to PE than Noah does and much faster drying rates overall. The differences between SMAP and Noah equivalent evaporation rates (Figure 6b and e) are smaller than the differences in drying rates because the top model layer thickness in Noah is twice SMAP's sensing depth.

The bottom panels of Figure 6 show that evaporative efficiency depends almost linearly on soil moisture and is not affected
by the PE rate, for both SMAP and Noah. At most VSM levels, the standard errors overlap between PE terciles. The evaporative efficiencies themselves are relatively low for both SMAP and Noah (Figure 7). Ninety percent of the values are lower than 0.27 and 0.21, respectively. We consider the implications of this result in the discussion. SMAP exhibits a considerable number of evaporative efficiency values below zero. These exist because noise in the SMAP observations leads to apparent wetting between successive overpasses during drying intervals, as can be seen for several cases in Figure 2.
Small precipitation events (that do not end the drydown) have a similar effect for both SMAP and Noah.

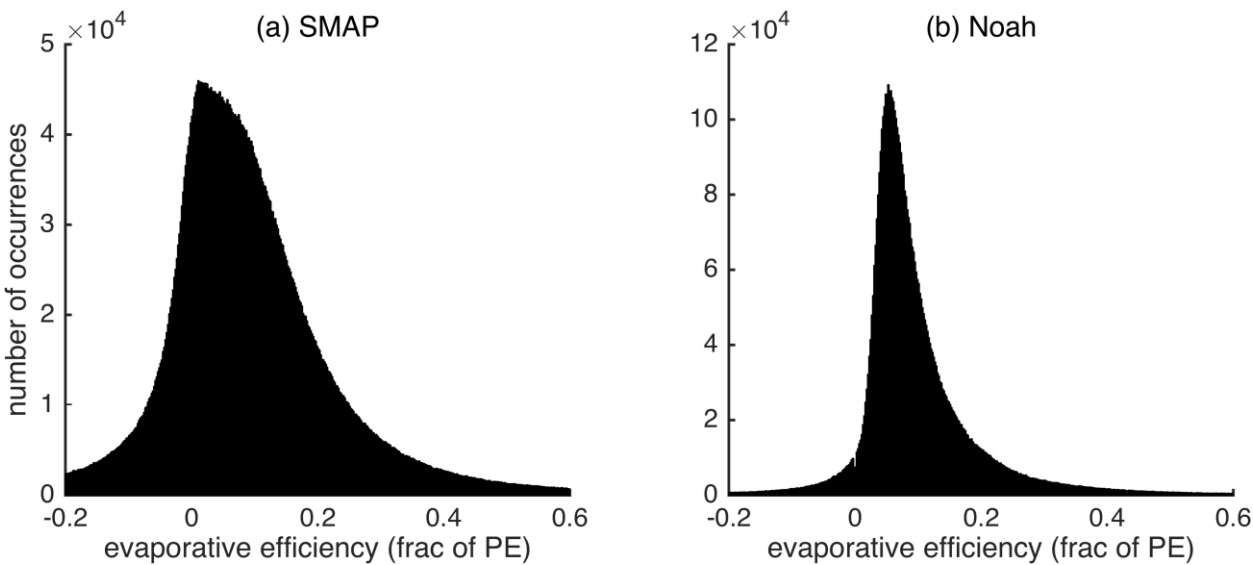

Figure 7: Frequency of evaporative efficiencies occurrences as observed by SMAP (a) and as simulated by Noah (b).

**3.2 Vegetation**

The influence of vegetation on drying rates is complex – vegetation may slow drying due to shading or increase it due to
transpiration from the near surface soil. An additional confounding factor is that vegetation tends to be more extensive in the summer months when PE is also high. A positive correlation exists between PE and NDVI across the analysis domain, with the exception of the lowest NDVI quantile (Figure 8). However, this domain-wide relationship masks strong regional



variations. In the humid upper midwest, PE and NDVI are strongly positively correlated. In contrast, NDVI is always low and PE varies greatly throughout the year in the desert southwest. Given that higher PE leads to higher drying rates (Sect. 3.1), we control for this effect by considering evaporative efficiency, which is normalized by PE.

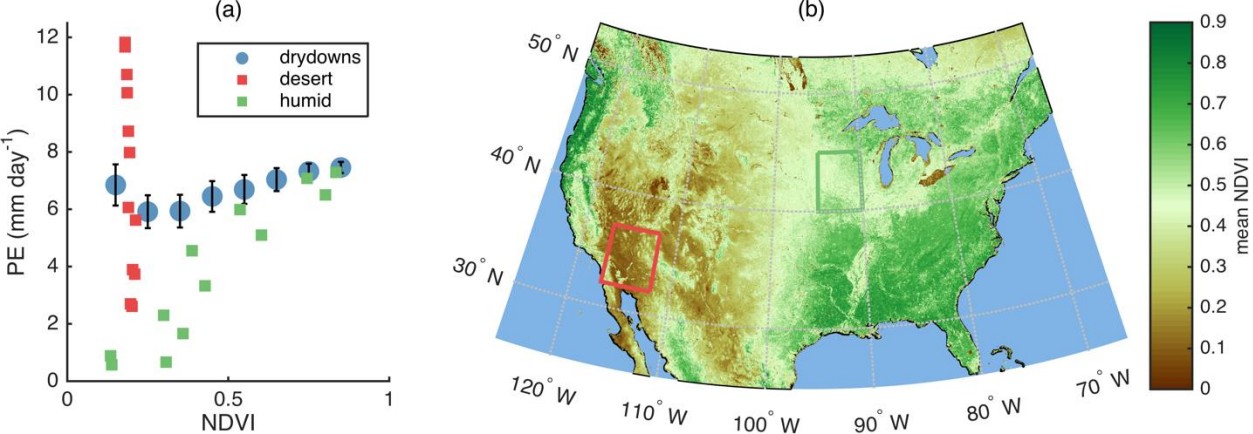

**Figure 8: (a) Circles show median PE as a function of 8 equal-width bins of increasing NDVI for all drydowns used in this study. Squares show average monthly relationships between PE and NDVI in two regions. (b) Extent indicators show locations of two regions.**

Figure 9 illustrates the relationship between vegetation amount (as indicated by NDVI) and drying rates, equivalent evaporation rates, and evaporative efficiencies. Corresponding SMAP and Noah observations are divided into 10 vegetation quantiles. The data are further divided into wet, transition, and dry soil (three quantiles), as observed by SMAP or simulated by Noah. Thus, each pair of corresponding SMAP and Noah values is included in the same vegetation quantile, but not necessarily in the same VSM quantile.







**Figure 9: As in Figure 6, but drying rates are a function of NDVI (8 quantiles, on x-axis) and surface soil moisture content (3 quantiles, by color).**





When the soil is dry, SMAP drying and equivalent evaporation rates are low regardless of vegetation level (Figure 9a and b), consistent with the results shown above (Figure 6). Similarly, the evaporative efficiency is very low for dry soil regardless of vegetation amount. For the intermediate soil moisture tercile, SMAP drying and evaporation rates both decrease as vegetation cover increases. The wettest tercile exhibits the lowest rates with intermediate amounts of vegetation cover, and

5  higher rates with both less and more vegetation. Once the drying rates are normalized by PE however, the relationship between soil drying and vegetation is more consistent across soil moisture levels (Figure 9c): evaporative efficiency consistently decreases as vegetation increases, unless the soil is dry and drying rates are effectively zero. Evaporative efficiency decreases by a factor of two to three between the lowest to the highest vegetation quantiles.

Noah simulations exhibit little or no relationship between vegetation and drying or evaporation rates (Figure 9d and e). For

10  the wettest soil tercile, evaporative efficiency does appear to decrease with vegetation cover, although the sensitivity is much less than found using SMAP observations.

### 3.3 Soil texture

The four main soil types in the CONUS (sand, sandy loam, loam, and silt loam) all exhibit similar drying dynamics as observed by SMAP. Figure 10a-c show that differences between the four texture classes are small: drying rates and

15  equivalent evaporation rates are slightly higher for loam, especially in wet soils, and slightly lower for silt loam, especially in soils of intermediate wetness (Figure 10a and b). Only minor differences exist between other texture types. The observed differences diminish slightly when drying is expressed as evaporative efficiency (Figure 10c), suggesting that some portion of the observed differences in drying and evaporation rates are due to spatial variations in PE that covary with soil texture.





**Figure 10: As in Figure 6, but drying rates are divided among the four most prevalent soil texture classes. Markers are omitted if fewer than 0.05 % of the total calculated drying rates fall within the VSM/soil texture category.**





In contrast, drying dynamics simulated by Noah (Figure 10d-f) exhibit large differences between the four most common texture classes. The coarsest texture (sand) shows the fastest drying rates, evaporation rates, and evaporative efficiencies. The finest texture (silt loam) shows the slowest. In wet soil conditions, the median evaporative efficiency for sand is approximately five times higher than that from other texture classes.

## 3.4 Inconsistencies between SMAP and Noah

The relationships between SMAP drying dynamics and related environmental factors are completely independent from the Noah model results. Therefore, the results presented here document how SMAP (alone) observes drying and its variations with key factors. The comparison between SMAP and Noah is not as straightforward. Two potential inconsistencies must be considered: the model parameterization of vegetation and the model layer depth.

## 3.4.1 NDVI vs $F_G$

The SMAP satellite observes varying land surface conditions. NDVI provides one estimate of vegetation status at the time of each SMAP retrieval. In contrast, the $F_G$ parameter used in Noah is based on the climatology of NDVI (Gutman and Ignatov, 1998). Therefore, the effect of vegetation on Noah model states is simplified: the model simulations cannot respond to deviations from climatology. Abnormally high or low vegetation cover that may exist in any year as indicated by NDVI (which was used for the analysis shown in Figure 9) will not affect the Noah simulation.

It is critical to evaluate if Noah's limited sensitivity to vegetation (Figure 9) is due to the mismatch between instantaneous (used in our analysis) and climatological (used in the Noah simulation) vegetation state. The alternative is that the lack of sensitivity to vegetation is attributable to the Noah model structure itself. In Figure 11, we show observed NDVI and Noah's climatological $F_G$ at six sites for the period of record of SMAP data analyzed here. Because the NDVI scaling parameters used to convert NDVI to $F_G$ (Eq. 1) are close to 0 and 1 [*Xia*, pers comm], the two variables can be plotted on the same axis. There are no clear departures from climatology. This suggests the Noah model structure is the source of the limited sensitivity of drying to vegetation amount, not the use of vegetation climatology. Furthermore, re-creating Figure 9 using $F_G$ as the covariate instead of NDVI does not substantially change the results (not shown).





**Figure 11: NDVI and $F_G$ at six sites in the study domain.**

### 3.4.2 Noah simulation Depth

Drying dynamics may be affected by the depth of soil being sensed or modeled (Rondinelli et al., 2015; Shellito et al., 2016b). We use the six supplementary simulations described in Sect. 2.1.5 to compare the drying characteristics of Noah





simulations from a 10 cm surface layer against a 5 cm surface layer. The former is the standard setup and was used for the results shown above. Figure 12 shows that the modeled dependence of drying rates on surface VSM is nearly identical between the simulations with different surface layer depths. Drying rates increase by only 6 % when the shallower layer is used. This slight change does shift Noah drying rates towards that documented by SMAP observations, but it is not large

5   enough to account for the approximately 4-fold differences between the two shown in Figure 6a and d. These results suggest that the differences in dynamics between SMAP and Noah are not attributable to the difference in depth between the two sources, but instead to the model's structure itself.

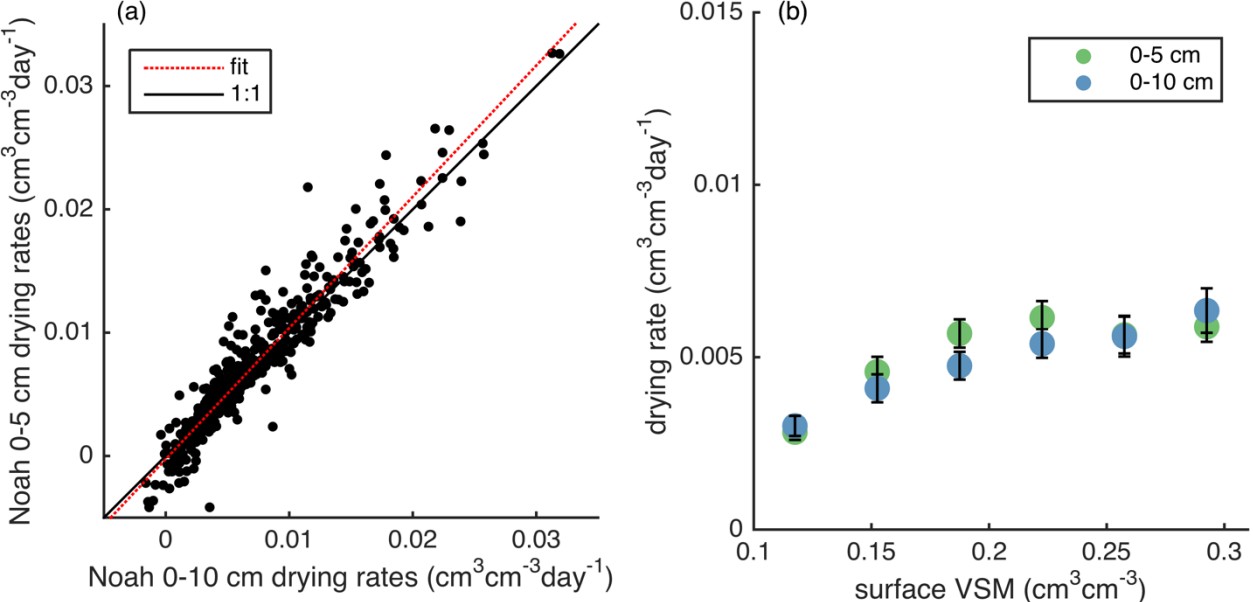

**Figure 12: (a) Scatter plot comparing Noah drying rates between simulations with different surface layer depths. R=0.93. Best fit intercept=0, slope=1.06. (b) Drying rate as a function of VSM for six quantiles using the two Noah simulation depths. Medians shown with markers, standard errors shown with error bars.**

## 4 Discussion

The remotely-sensed data from NASA's SMAP mission, combined with modeled data, provide insight into the environmental factors that affect surface soil moisture dynamics and direct evaporation from soil. The results presented here

15  are based on correlations between environmental factors and soil drying rates, so it is not possible to prove cause–effect relationships exist. However, the results are consistent with ground-based observations (e.g., Kurc and Small, 2004) and physics-based relationships included in models (e.g., Laio et al., 2001). Therefore, results confirm that these fundamental relationships exist at the continental scale.

SMAP data show that the land surface dries rapidly immediately after rainfall. With time, the soil dries and the drying rate

20  slows, the latter approaching zero after ~10 days. Noah simulations also exhibit this trend, but the drying rates are slower




directly after rainfall and persist at non-zero values for longer than SMAP rates do, indicating a more linear drying process (Figure 5).

To constrain the factors affecting the drying process, we simultaneously consider the supply of water (soil moisture) and the atmospheric demand for it (PE). When the surface soil is wet, only atmospheric demand should limit evaporation rates (stage one evaporation). When the surface soil is dry, moisture supply further restricts evaporation rates (stage two evaporation). Our study finds that, at SMAP spatial and temporal scales, the satellite is observing a system that is predominantly water-limited (McColl et al., 2017a). This is supported by the following observations (Figure 6):

(1) In most cases, drying and evaporation rates are linearly related to soil moisture content. Such consistent dependence on moisture supply indicates water-limited conditions. In contrast, energy-limited conditions would show evaporation rates to be insensitive to soil moisture.

(2) The sensitively of drying rates to soil moisture depends on PE. This, along with Figure 4, which shows an approximately 1:1 ratio of drying rates to evaporation rates, supports the supposition that drying rates are controlled mainly by evaporation rates instead of by drainage or diffusion rates, as they would be in an energy-limited environment.

(3) In the low PE tercile, there is a plateau in drying rates when soil moisture exceeds 0.15 $cm^3$ $cm^{-3}$ (Figure 6a). Such a plateau could indicate an energy-limited environment. However, the evaporative efficiency does not also reach a plateau at high VSM. This suggests variations in PE within the lowest PE tercile are responsible for the plateau observed in Figure 6a and b – the wettest soils are found in environments with the very lowest PE rates.

Noah simulations are consistent with the results from SMAP data. In general, drying rates from Noah are much lower than those from SMAP, but the equivalent evaporation rates from Noah are the same as or higher than those from SMAP (Figure 5c) because the layer 1 depth is greater and soil moisture levels are generally higher in Noah (Figure 5a). Drying rates do not plateau in the low PE tercile, further supporting the idea that the system is water-limited.

The calculated values for evaporative efficiency are quite low (Figure 7); 90 % of values are below ~0.25. This suggests evaporation from the surface satisfies only a small fraction of the atmospheric demand. The fraction is not larger for one or more of the following reasons: (1) at this spatial and temporal scale, evaporation is highly water-limited, (2) most plant roots are deeper than ~5 cm, so transpiration draws water from depths below the surface layer, and thus does not contribute to the SMAP-based accounting of drying shown here, (3) similarly, if any evaporation draws moisture from below SMAP's sensing depth, it will not be accounted for here, and (4) the sampling interval of SMAP is too low to capture the very fastest evaporation rates, which occur soon after rainfall. Only 6.5 % of the calculated drying rates include an observation from within the first 12 hours after precipitation, when the fastest drying and highest evaporative efficiency is likely to occur. For example, in Figure 3, the first observation, $t_1$, is almost two days after rainfall cessation.

While VSM and PE clearly influence soil drying, SMAP observations also show that vegetation cover plays an important role in determining soil drying rates. These effects are most obvious once drying rates are normalized by PE, as the highest





NDVI, and therefore vegetation amount, occur during the summer months when PE tends to be greatest (Figure 8). Evaporative efficiency decreases with increasing vegetation cover (Figure 9c), unless the soil is dry and evaporative efficiency is close to zero. This means that at a given moisture level and PE rate, the surface soil of a parcel of land will dry more slowly if it has vegetation on it than if it does not. This is consistent with ground-based observations that show direct

evaporation can be limited by vegetation (e.g., Breshears et al., 1998).

In comparison to SMAP, the effects of vegetation on drying in Noah are minimal (Figure 9e-f). These results can be understood given the model's formulation of evapotranspiration (Chen and Dudhia, 2001). Direct soil evaporation only occurs over the fraction of land surface not shaded by the canopy ($1$-$F_G$), so evaporation decreases with more vegetation. Transpiration only occurs over the fraction of land surface that has a canopy ($F_G$), so transpiration increases with more

vegetation. Our results show these two effects are balanced in dry and intermediate wetness soils. Soil in the wettest tercile exhibit slightly higher evaporative efficiencies from bare soil than from vegetated soil (Figure 9f). The Noah results indicate that in the top 10 cm of soil, moisture from rainfall enhances evaporation from bare ground more than it does transpiration from vegetated ground.

Vegetation effects are much greater as observed by SMAP than simulated by Noah. Therefore, the direct evaporation flux in

Noah should be greater (at a given VSM and PE), which would result in higher surface drying efficiencies when vegetation is sparse. *Betts et al.,* (1997) and *Ek et al.,* (2003) both modified Noah's bare soil evaporation function to magnify the decrease in evaporation as the surface dries. The SMAP observations suggest further adjustments are needed.

The results in Figure 10 show that the sensitivity to soil texture is too high in Noah. SMAP shows only small differences in drying dynamics related to texture classes (here and McColl et al., 2017a), whereas Noah simulations indicate variations of a

factor of five between sand and other texture classes. Noah-simulated results conform to the expectation that coarser soils (sand and sandy loam) dry faster than fine-grained soils (silt loam). (Soil infiltration and redistribution parameters are indeed selected according to texture class (Chen and Dudhia, 2001)). On the contrary, our SMAP-based results show the role of soil textures to be less important than the other factors analyzed here.

The differences in behavior between SMAP and Noah can also partially be attributed to differences in sensing and

simulation depths. We expect thicker surface layer dynamics to be dampened when compared to a thinner layer (e.g., Rondinelli et al., 2015). However, changing Noah's layer 1 soil depth from 0-10 cm to 0-5 cm only increases soil drying rates by 6 % (Figure 12), implying that the model structure itself prevents Noah from accurately reproducing the surface soil moisture dynamics observed by SMAP.

## 5 Conclusion

(1) SMAP-observed and Noah-simulated soil moisture and drying rates decrease with time since precipitation. SMAP drying rates are faster than Noah-simulated drying rates in the first 8 days after rainfall, but slower



afterwards. Because Noah's top soil layer is twice the depth that SMAP senses, its equivalent evaporation rates are nearly the same as SMAP's soon after precipitation and higher afterwards.

(2) SMAP-observed and Noah-simulated soil drying rates both vary linearly with soil moisture content, evidence that continental-scale soil moisture dynamics operate in a water-limited system.

(3) Equivalent evaporation rates from SMAP and Noah rarely exceed 1 mm day$^{-1}$. Expressed as evaporative efficiency, 90 % of the calculated rates fall below 0.27 (SMAP) or 0.21 (Noah). These values are far below unity, providing further evidence of a water-limited environment. However, extraction of water by transpiration from below the surface soil would shift total ET efficiencies closer to 1.

(4) SMAP and Noah both show that high atmospheric demand for moisture (high PE) increases the sensitivity of drying rates to soil moisture content.

(5) More vegetation amount, indicated by higher NDVI, decreases the surface drying efficiency: SMAP shows a 3-fold evaporation efficiency decrease between sparsely-vegetated and densely-vegetated pixels. This suggests that the decreases in evaporation from canopy shading are not offset by increases in transpiration from the shallow soil layer. Noah shows a much smaller decrease in evaporative efficiency, only for wet soils, suggesting a deficiency in the model structure.

(6) Soil texture class has a small influence on SMAP drying dynamics. Noah drying dynamics are strongly affected by soil texture class, as prescribed by its soil hydraulic property parameterization.

**Data availability**

The SMAP retrievals (O'Neill et al., 2016), NLDAS-2 forcings (Xia et al., 2009), NLDAS-2 Noah simulations (Xia et al., 2012), and MODIS NDVI data (NASA, 2016) used in this study can be obtained from public repositories.

**Competing interests**

The authors declare that they have no conflict of interest.

**Acknowledgements**

The authors would like to thank Dr. Andrew Badger for downloading and processing MODIS NDVI data.

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
