# Peer review of "Controls on surface soil drying rates observed by SMAP and simulated by the Noah land surface model"

_Hydrology and Earth System Sciences, 2017_

## Referee Comment (RC1) · Anonymous Referee #1 · 22 Aug 2017

The authors estimate soil drying rates by identifying soil moisture "drydowns" in time series of SMAP observations and Noah land surface model outputs. They compare the estimated drydown rates to covariates such as NDVI, potential evaporation and soil texture, and discuss differences between SMAP and Noah. The study is fairly well written, and the analyses are generally thorough. For example, the authors thoroughly consider important factors such as differences between soil moisture depths of SMAP and Noah in their analyses.

The paper's biggest weakness, in my view, is in its motivation and framing. The story should be tighter. For example, the first line of the abstract says "Drydown periods that

follow precipitation events provide an opportunity to assess the mechanisms by which soil moisture dissipates from the land surface." Various mechanisms contribute to soil moisture dissipation – bare soil evaporation, transpiration, percolation, runoff – but only one of these mechanisms is directly estimated from drydowns in the manuscript (bare soil evaporation). It is not clear to me how one would assess the other mechanisms by examining drydowns, and the reader is not provided with any further guidance or results in the manuscript. Perhaps the story could be reframed around estimating bare soil evaporation.

As far as I can tell, this is essentially a model validation study, except it excludes much of the available data from the analysis (i.e., only includes drydowns). What is gained from doing this that could not be gained from a standard validation study that uses the full time series of data? The authors partially and indirectly answer this question, but the paper would be more compelling if they spent more time directly addressing the paper's motivation. What exactly do we learn from examining soil moisture drydowns in isolation? What specific additional information do we obtain from this analysis compared to a standard validation study that uses all the data, not just drydowns?

The lack of a clear story also leads to some strange decisions in the data analysis. In Figure 5, for example, soil moisture drydowns are averaged across space and time, but with only a partial normalization in time, and no normalization in space. Averaging nonlinear (for example, exponential) drydowns with varying additive and multiplicative biases would be expected to significantly dampen the nonlinearity and cloud interpretation. Removing the mean of each drydown and scaling VSM to a saturation ratio before averaging would better preserve the expected exponential form of the average drydown. The authors' data analysis choices are not necessarily wrong, but they need to be justified in the context of a larger story that is currently not well-articulated.

I am also concerned about the robustness of the approach used by the authors to estimate drydowns, which relatively frequently produces unphysical, negative evaporative efficiencies. I offer some specific suggestions below.

[Figure]

Specific comments The introduction could be tighter. A lot of space is devoted to recapping fundamental physics of vadose zone hydrology. Much of this could be edited down, moved to later parts of the manuscript, or cut completely. Focus on motivation for the study here.

p. 3, line 29: "Soil moisture supply, PE rate, and vegetation cover are observed..." PE rate is not observed. It is estimated from a reanalysis which is significantly model-based.

Fig. 1: The SMAP orbit is clearly visible in Figs 1a and 1b. This should be noted as an unphysical artifact of the observations.

p. 7, line 3: how sensitive are the results to choices made in the automated selection process? Comment on this in the manuscript.

p. 8, line 9: "This is an artifact of the SMAP algorithm and does not reflect the drying process." How do the authors know this? Please cite a reference or personal communication, if necessary.

p. 8, line 21: "Noah drying rates have a nearly 1:1 relationship with evaporation rates." The relationship is substantially noisy with R = 0.47, and clearly not "1:1". Perhaps rephrase to something like "the slope of the regression line is nearly 1".

p. 14, line 12: "SMAP exhibits a considerable number of evaporative efficiency values below zero." Indeed, there is a substantial fraction below zero, and this makes me question the accuracy of the estimates above zero, too. The authors attribute this to noise in the SMAP observations; if it is due to noise, then the authors need to redesign their analyses to be more robust to it. The authors should revisit their drydown algorithm given on p. 7 to ensure this fraction is lower. For instance, they could return to fitting an exponential model, rather than directly estimating soil drying rates from finite differences. They should also alter the criterion "the dry period must be at least 3 days long" (p. 7) to require the dry period to be longer. Three days is quite short given

the SMAP revisit time is ∼3 days and makes the algorithm highly susceptible to noise in the observations.

p. 15, line 8: "vegetation amount (as indicated by NDVI)" NDVI is a measure of greenness, not vegetation amount. This is only loosely correlated with vegetation amount. Please add nuance to this statement. Also, since NDVI is an optical index and is therefore not observed during cloudy conditions, how do the authors expect this to impact their results? For example, cloudy conditions will lower PE, all else being equal; therefore these conditions are being systematically excluded from the comparison. Comment on this in the manuscript.

p. 19, line 6: "…are completely independent from the Noah model results." NDVI and FG are hardly independent of one another: FG is estimated directly from NDVI!

p. 21, line 17: "Therefore, results confirm that these fundamental relationships exist at the continental scale." To me, this is the most interesting story in the manuscript: translating known results from the point scale, to continental scales.

p. 23, line 1: "NDVI, and therefore vegetation amount,…" NDVI is a measure of greenness, not vegetation amount.

p. 23, line 18: "The results in Figure 10 show that the sensitivity to soil texture is too high in Noah." Possibly, although an alternative explanation could be that the soil texture maps used are themselves error prone and insufficient, particularly at the large spatial scales relevant to SMAP and Noah.

---

## Referee Comment (RC2) · Anonymous Referee #2 · 9 Sep 2017

General Comments:

This article calculates drying rates over parts of North America and assesses the relative roles of other land surface characteristics such as vegetation and soil texture in soil dry down. This paper extends previous work by expressing soil dry down in multiple ways and by comparing to Noah land surface model simulations. A key finding is that SMAP dries down more quickly after precipitation than Noah and that evaporative efficiency is reduced when vegetation is increased.

The article is well-written and will certainly be of interest to the land community. The methods and the results presented here are useful from a soil science perspective

but also seem likely to be helpful in better understanding and improving land surface models. I offer minor comments below to improve the readability of the manuscript.

Specific Comments:

Page 1, Line 10-11: "Data cover the domain of the NLDAS 2". Please reword – just because the data cover this domain, doesn't meant that the whole domain will be used (as is the case here).

Page 3 Line 6 – Although the nominal SMAP depth is 5 cm, the sensing depth also changes slightly with moisture content. Is it possible that this could affect the conclusions?

Page 6 Line 24: Was any information lost in the re-gridding? How do you know (comparison of statistics to the original, etc)?

Page 10, Section 2.2.5: This section is very short and doesn't seem substantial enough to be its own section. Perhaps move the information to 2.1.5.

Page 14 line 14: I don't see the wetting between successive overpasses in Figure 2. Is it possible to point out a time period as an example?

Figure 4: Is it possible to also include the points on this plot, rather than just the contours?

Technical Corrections:

Page 4, Line 6-7: Please write out the words first and have the abbreviations in parentheses.

Page 4, Line 13-14: (cm3 cm-3) instead of ", in cm3 cm-3"

Page 6 Line 7: "on it" not necessary and sounds a bit awkward.

Page 6, Line 9: The equations appear a bit fuzzy. Is it possible to make these clearer?

Page 6, Line 16: I believe the cities should be separated with semi-colons, rather than

commas (e.g., Fort Cobb, OK; Little River, GA; . . .etc).

Page 8, Line 13: covert to convert

---

## Author Comment (AC1) · 16 Oct 2017

**Final Author Comments**

We thank the two anonymous referees for their helpful and productive comments. We address the comments of the first, followed by the second. For each, we divide comments into general comments and specific comments, and our responses are provided directly after each.
* * *
**Referee One:**

**General Comment 1:**

The authors estimate soil drying rates by identifying soil moisture "drydowns" in time series of SMAP observations and Noah land surface model outputs. They compare the estimated drydown rates to covariates such as NDVI, potential evaporation and soil texture, and discuss differences between SMAP and Noah. The study is fairly well written, and the analyses are generally thorough. For example, the authors thoroughly consider important factors such as differences between soil moisture depths of SMAP and Noah in their analyses.

The paper's biggest weakness, in my view, is in its motivation and framing. The story should be tighter. For example, the first line of the abstract says "Drydown periods that follow precipitation events provide an opportunity to assess the mechanisms by which soil moisture dissipates from the land surface." Various mechanisms contribute to soil moisture dissipation – bare soil evaporation, transpiration, percolation, runoff – but only one of these mechanisms is directly estimated from drydowns in the manuscript (bare soil evaporation). It is not clear to me how one would assess the other mechanisms by examining drydowns, and the reader is not provided with any further guidance or results in the manuscript. Perhaps the story could be reframed around estimating bare soil evaporation.

**General Response 1:**

SMAP data can tell us more about bare soil evaporation than other fluxes (transpiration, percolation, and runoff) because (1) the timescale of SMAP observations does not allow for measurements of soil drying until 1 to several days after rainfall -- thus drainage and redistribution have largely ceased (see McColl et al., 2017 for a detailed analysis); and (2) the depth over which the satellite is sensitive to soil moisture necessarily focuses our study on the top few centimeters, where evaporation affects moisture more than transpiration does. Transpiration extracts soil moisture from throughout the entire root zone, usually ~1m thick. These introductory items were presented in the manuscript on pg 3, lines 3-9, but we will enhance the discussion and be more explicit. We will clarify this in the revised manuscript by moving pg 3, lines 3-9 closer to the beginning of the introduction and omitting some of the text dedicated to summarizing vadose zone hydrology (pg 2, lines 3-13). The linkage to the "loss analysis" in McColl et al. (2017) will be strengthened.

Reference:
McColl, K. A., Wang, W., Peng, B., Akbar, R., Short Gianotti, D. J., Lu, H., Pan, M. and Entekhabi, D.: Global characterization of surface soil moisture drydowns, Geophys. Res. Lett., doi:10.1002/2017GL072819, 2017.

**General Comment 2:**

As far as I can tell, this is essentially a model validation study, except it excludes much of the available data from the analysis (i.e., only includes drydowns). What is gained from doing this that could not be gained from a standard validation study that uses the full time series of data? The authors partially and indirectly answer this question, but the paper would be more compelling if they spent more time directly addressing the paper's motivation. What exactly do we learn from examining soil moisture drydowns in isolation? What specific additional information do we obtain from this analysis compared to a standard validation study that uses all the data, not just drydowns?

**General Response 2:**

Our primary goal is to use SMAP data to investigate the processes governing soil drying at a large scale, and then use the resulting information to evaluate some particular aspects of Noah LSM performance. It is certainly important (and feasible) to utilize SMAP to quantify the rate and magnitude of soil wetting during precipitation. We are aware of ongoing studies using soil wetting from SMAP to identify problems with precipitation datasets. This is beyond the scope of our study.

Soil drying is a complex process that is of great interest to the hydrologic community (pg 1, lines 23-25). The timescale of soil drying affects the fluxes of water, energy, and carbon between the land surface and atmosphere – which is a fundamental aspect of weather and climate documented in many previous studies. Soil drying alone (not wetting) has been the focus of at least three recent studies using SMAP data (Shellito et al., 2016; McColl et al., 2017; McColl et al., 2017b), although the analyses and results described here are very different from the previous papers.

We include Noah LSM in our drying analysis because it allows us to examine the soil drying process in the model. A "standard validation" study would not permit for such a focused analysis of one (critical) aspect of a relatively complex model.

We will clarify these objectives in the introduction of the revised manuscript, including (1) the importance of soil drying on land-atmosphere interactions; and (2) how this analysis is useful to probe one component of a commonly-used LSM.

References:

McColl, K. A., Wang, W., Peng, B., Akbar, R., Short Gianotti, D. J., Lu, H., Pan, M. and Entekhabi, D.: Global characterization of surface soil moisture drydowns, Geophys. Res. Lett., doi:10.1002/2017GL072819, 2017.

McColl, K. A., Alemohammad, S. H., Akbar, R., Konings, A. G., Yueh, S. and Entekhabi, D.: The global distribution and dynamics of surface soil moisture, Nat. Geosci., doi:10.1038/ngeo2868, 2017b.

Shellito, P. J., Small, E. E., Colliander, A., Bindlish, R., Cosh, M. H., Berg, A. A., Bosch, D. D., Caldwell, T. G., Goodrich, D. C., McNairn, H., Prueger, J. H., Starks, P. J., van der Velde, R. and Walker, J. P.: SMAP soil moisture drying more rapid than observed in situ following rainfall events, Geophys. Res. Lett., 43(15), 8068–8075, doi:10.1002/2016GL069946, 2016.

**General Comment 3:**

The lack of a clear story also leads to some strange decisions in the data analysis. In Figure 5, for example, soil moisture drydowns are averaged across space and time, but with only a partial normalization in time, and no normalization in space. Averaging nonlinear (for example,

exponential) drydowns with varying additive and multiplicative biases would be expected to significantly dampen the nonlinearity and cloud interpretation. Removing the mean of each drydown and scaling VSM to a saturation ratio before averaging would better preserve the expected exponential form of the average drydown. The authors' data analysis choices are not necessarily wrong, but they need to be justified in the context of a larger story that is currently not well-articulated.

**General Response 3:**

The concerns of this reviewer regarding data analysis choices are perhaps caused by a misunderstanding of some of the figures. Specifically, the referee mentions that "In Figure 5, for example, soil moisture drydowns are averaged across space and time…" and expands on how such averaging could cloud interpretation. Figure 5 (and 6 and 9) are not showing averages, but rather median values from the SMAP data binned according to various criteria. Although we did describe this in the original manuscript (e.g., figure caption for figure 5, text on Page 9, L19), the review comment shows the point was not made clearly enough. We will expand on the method used and remind the reader in the results section that median values were used.

The suggestion of "scaling VSM to a saturation ratio," however, is undesirable for our purposes. The calculation of a saturation value would require an estimate of soil texture. Continental-scale soil texture maps are unreliable and this paper shows that Noah's dependence on them is too great (pg 23, line 18).

**General Comment 4:**

I am also concerned about the robustness of the approach used by the authors to estimate drydowns, which relatively frequently produces unphysical, negative evaporative efficiencies. I offer some specific suggestions below.

p. 14, line 12: "SMAP exhibits a considerable number of evaporative efficiency values below zero." Indeed, there is a substantial fraction below zero, and this makes me question the accuracy of the estimates above zero, too. The authors attribute this to noise in the SMAP observations; if it is due to noise, then the authors need to redesign their analyses to be more robust to it. The authors should revisit their drydown algorithm given on p. 7 to ensure this fraction is lower. For instance, they could return to fitting an exponential model, rather than directly estimating soil drying rates from finite differences. They should also alter the criterion "the dry period must be at least 3 days long" (p. 7) to require the dry period to be longer. Three days is quite short given the SMAP revisit time is ~3 days and makes the algorithm highly susceptible to noise in the observations.

**General Response 4:**

SMAP evaporative efficiency values below zero can be attributed to (1) the expected noise in SMAP observations (up to 0.04 $cm^3$ $cm^{-3}$), (2) that soil drying rates trend towards zero in a given dry period (thereby increasing the role of the aforementioned noise), and (3) small amounts of rainfall (that are not large enough to trigger the end of a drydown period or that are not included in the NLDAS forcings) can cause real increases in soil moisture – which appear as 'negative evaporation' when included in the drydown period.

To account for these error sources, we can flag data that exhibit changes of less than 0.04 $cm^3$ $cm^{-3}$, data that constitute the driest 10% of SMAP observations (moisture values below 0.05 $cm^3$ $cm^{-3}$), and data that span even light precipitation. Removing data with those three flags individually results in a decrease in the number of nonphysical evaporative efficiency values (from 23% to 10%, to 20%, and to 21%, respectively). Removing all three types of flagged data

at once results in mere 6.6% of SMAP evaporative efficiencies below zero. We will include this analysis in the revised paper and move Figure 7 to an earlier position in the manuscript. Additionally, we will overlay two subsets of the population onto the Figure 7 histogram: data that have high soil moisture (and thus will have higher drying rates) and data that have low soil moisture (and thus will have low and sometimes negative drying rates). These subsets can then be more effectively linked to the data shown in Figure 6c, making it clearer that the data falling below zero are attributable to noise in the SMAP retrievals.

While drying values close to or below zero could be excluded from our analysis, it would not change any of the results or conclusions of the analysis. Even if there are noise in the data (within the mission target specified by SMAP), extremely clear relationships exist between soil drying and geophysical variables (Fig 5, 6, and 9). For example, Figure 6a shows clear differences in drying behavior even at values below $0.02$ cm$^3$ cm$^{-3}$ day$^{-1}$ (SMAP's noise target for the case of a 2-day overpass). This shows there is value in including SMAP drying increments on the low end of the spectrum. Even with the noise inherent in the ***individual*** observations (i.e., some negative evaporative efficiencies), the large volume of data available provides clear and statistically significant results. Only at the driest soil levels ($0.05$ cm$^3$ cm$^{-3}$) do the standard errors in Figure 6a (and b and c) extend below $0$ cm$^3$ cm$^{-3}$ day$^{-1}$.

Alternative methods of selecting drydowns, such as that used in McColl et al. (2017) or Shellito et al. (2016), include the exact same noise in SMAP observations – the noise just contributes to misfit in the exponential curve fit, rather than negative evaporation values from individual observations. Shellito et al. (2016) found nearly identical results using the exponential fit and finite differences, thus there is no justification for claiming one method is better than the other. In the revision, we will include a paragraph as to the differences between the two approaches, focusing on how noise affects results from each method.

Finally, because each drydown observation is calculated via finite differences, the length of the drydown period is unrelated to the number of "noisy" observations; only two observation go into each calculation, and SMAP overpasses occur every 1 to 3 days. Therefore, drydown periods of only 3 days are still useful and do not disproportionately contribute to noisy observations. We will clarify this point in the methods.

References:

McColl, K. A., Wang, W., Peng, B., Akbar, R., Short Gianotti, D. J., Lu, H., Pan, M. and Entekhabi, D.: Global characterization of surface soil moisture drydowns, Geophys. Res. Lett., doi:10.1002/2017GL072819, 2017.

Shellito, P. J., Small, E. E., Colliander, A., Bindlish, R., Cosh, M. H., Berg, A. A., Bosch, D. D., Caldwell, T. G., Goodrich, D. C., McNairn, H., Prueger, J. H., Starks, P. J., van der Velde, R. and Walker, J. P.: SMAP soil moisture drying more rapid than observed in situ following rainfall events, Geophys. Res. Lett., 43(15), 8068–8075, doi:10.1002/2016GL069946, 2016.

**Specific Comment (SC) 1:**

p. 3, line 29: "Soil moisture supply, PE rate, and vegetation cover are observed. . ." PE rate is not observed. It is estimated from a reanalysis which is significantly model-Based.

**Specific Response (SR) 1:**

We propose changing to "Soil moisture supply, PE rate, and vegetation cover are observed or calculated …"

**SC 2:**

Fig. 1: The SMAP orbit is clearly visible in Figs 1a and 1b. This should be noted as an unphysical artifact of the observations.

**SR 2:**

We will add a note to this effect in the caption.

**SC 3:**

p. 7, line 3: how sensitive are the results to choices made in the automated selection process? Comment on this in the manuscript.

**SR 3:**

Results are insensitive to minor changes made to the automated selection process. A sentence to this effect will be added.

**SC 4:**

p. 8, line 9: "This is an artifact of the SMAP algorithm and does not reflect the drying process." How do the authors know this? Please cite a reference or personal communication, if necessary.

**SR 4:**

We will cite A. Colliander, pers comm. Repeated high soil moisture levels is a feature of the retrieval algorithm when actual moisture levels are above those normally observed (~0.5).

**SC 5:**

p. 8, line 21: "Noah drying rates have a nearly 1:1 relationship with evaporation rates." The relationship is substantially noisy with R = 0.47, and clearly not "1:1". Perhaps rephrase to something like "the slope of the regression line is nearly 1".

**SR 5:**

We will change this wording accordingly.

**SC 6:**

[Specific comment 6 was copied into General Comment 4 and responded to in General Response 4].

**SC 7:**

p. 15, line 8: "vegetation amount (as indicated by NDVI)" NDVI is a measure of greenness, not vegetation amount. This is only loosely correlated with vegetation amount. Please add nuance to this statement. Also, since NDVI is an optical index and is therefore not observed during cloudy conditions, how do the authors expect this to impact their results? For example, cloudy conditions will lower PE, all else being equal; therefore these conditions are being systematically excluded from the comparison. Comment on this in the manuscript.

**SR 7:**

The NDVI product we use is a 16-day product. Each individual day's value is an interpolation between non-cloudy conditions. (See pg 5, lines 18-20.) Therefore, cloudy conditions do not affect the results of this study.

**SC 8:**

p. 19, line 6: ". . .are completely independent from the Noah model results." NDVI and FG are hardly independent of one another: FG is estimated directly from NDVI!

**SR 8:**

Thank you for this comment. This section will be re-written. Our intent is to point out that SMAP results come from observations and an algorithm that does not incorporate Noah simulations. In contrast, Noah simulations ingest Fg, which (as the referee points out) have been estimated directly from NDVI.

An alternative to pg 19, lines 6-9 is provided here:

"SMAP drying dynamics are dependent on environmental states observed at the time of each overpass. Noah, however, depends on a vegetation parameter that has been calculated from climatology: its $F_G$ parameter is derived from 5 years of NDVI observations (Gutman and Ignatov, 1998). In addition, Noah results reflect the behavior of the 10 cm surface layer, whereas SMAP nominally senses only the top 5 cm. These two inconsistencies must be considered carefully to fairly compare SMAP and Noah results."

Reference:

Gutman, G. and Ignatov, A.: The derivation of the green vegetation fraction from NOAA/AVHRR data for use in numerical weather prediction models, Int. J. Remote Sens., 19(8), 1533–1543, doi:10.1080/014311698215333, 1998

**SC 9:**

p. 21, line 17: "Therefore, results confirm that these fundamental relationships exist at the continental scale." To me, this is the most interesting story in the manuscript: translating known results from the point scale, to continental scales.

**SR 9:**

Thank you. We will be sure to keep this statement in a prominent location.

**SC 10:**

p. 23, line 1: "NDVI, and therefore vegetation amount. . ." NDVI is a measure of greenness, not vegetation amount.

**SR 10:**

Both NDVI and vegetation amount are expected to be greater in the summer months. To clarify that NDVI is not equal to vegetation amount, but merely an indicator of it, we propose the following change to the lines between pg 22 line 32 and pg 23 line 1: "While VSM and PE clearly influence soil drying, SMAP observations also show that vegetation plays an important role in determining soil drying rates. These effects are most obvious once drying rates are normalized by PE, because vegetation (as indicated by NDVI) and PE both tend to be greatest in the summer months (Figure 8)."

**SC 11:**

p. 23, line 18: "The results in Figure 10 show that the sensitivity to soil texture is too high in Noah." Possibly, although an alternative explanation could be that the soil texture maps used are themselves error prone and insufficient, particularly at the large spatial scales relevant to SMAP and Noah.

**SR 11:**

We can add a statement to include this possibility at the end of pg 23, line 23: "It is possible that improved soil texture maps could bring SMAP and Noah results into closer agreement, but the

heterogeneity of soils even within a single texture class (Gutmann and Small 2005) makes it unlikely that the solution will come from improving texture maps alone."
* * *
**Referee Two:**

**General Comment:**

This article calculates drying rates over parts of North America and assesses the relative roles of other land surface characteristics such as vegetation and soil texture in soil dry down. This paper extends previous work by expressing soil dry down in multiple ways and by comparing to Noah land surface model simulations. A key finding is that SMAP dries down more quickly after precipitation than Noah and that evaporative efficiency is reduced when vegetation is increased.

The article is well-written and will certainly be of interest to the land community. The methods and the results presented here are useful from a soil science perspective but also seem likely to be helpful in better understanding and improving land surface models. I offer minor comments below to improve the readability of the manuscript.

**Specific Comment (SC) 1:**
Page 1, Line 10-11: "Data cover the domain of the NLDAS 2". Please reword – just because the data cover this domain, doesn't meant that the whole domain will be used (as is the case here).

**Specific Response (SR) 1:**
The domain used in this study is in fact the NDLAS-2 domain.

**SC 2:**
Page 3 Line 6 – Although the nominal SMAP depth is 5 cm, the sensing depth also changes slightly with moisture content. Is it possible that this could affect the conclusions?

**SR 2:**
Yes it is possible that the varying sensing depth could affect the conclusions. We will add a sentence on pg 3, line 7: "...will propagate down to greater depths. In addition, L-band sensing depth varies slightly with moisture content (Njoku and Kong 1977), though quantifying the effect of this change is beyond the scope of this study." In addition, we will add the word "nominally" to pg 4, line 14 to clarify that this is not an exact sensing depth. In the discussion, we will add the following between the sentences on pg 22, line 22: "In addition, it is possible that SMAP drying rates are exaggerated due to slight decreases in L-band sensing depth that accompany wet soil (Njoku and Kong 1977). After rainfall, moisture in the top couple centimeters could dominate the signal, leading to the entire 0-5 cm sensing depth being assigned a moisture level that is only present at the very surface. As the soil dries and becomes more evenly distributed within the sensing depth, such abnormalities would dissipate."

**SC 3:**
Page 6 Line 24: Was any information lost in the re-gridding? How do you know (comparison of statistics to the original, etc)?

**SR 3:**

The analysis in this study would not have been possible without some sort of assignation of MODIS and NLDAS-2 pixels into the SMAP EASE-2 grid (regridding). The native grid for NDVI is finer than the SMAP grid, so there is concern for loss of information in that conversion. To that end, we compare empirical PDFs of average annual NDVI before and after re-gridding. The native and re-gridded data are nearly indistinguishable from one another (Figure R1).

The native NLDAS-2 grid is slightly coarser than the SMAP grid. This means that the conversion did not lose any data, but rather that information was occasionally repeated, as described in pg 6, lines 27-30: "The NLDAS-2 grid is only slightly coarser than SMAP's grid, so occasionally the same data will be mapped into two SMAP pixels. Though this is not ideal, it is preferable to basing our analysis on the NLDAS-2 grid, which would force us to exclude some SMAP pixels or blend them with their neighbor when they fall within the same NLDAS-2 pixel."

[Figure]

Figure R1: Empirical probability distribution functions of native and re-gridded NDVI data.

**SC 4:**

Page 10, Section 2.2.5: This section is very short and doesn't seem substantial enough to be its own section. Perhaps move the information to 2.1.5.

**SR 4:**

Thank you for this suggestion. The change will be easy and increase readability.

**SC 5:**

Page 14 line 14: I don't see the wetting between successive overpasses in Figure 2. Is it possible to point out a time period as an example?

**SR 5:**

Add to page 14, line 14 "... as can be seen for several cases in the upper panels of Figure 2: late November at Fort Cobb, OK, and mid October at Marena, OK."

**SC 6:**

Figure 4: Is it possible to also include the points on this plot, rather than just the contours?

**SR 6:**

We have recreated Figure 4 with 1 in 1000 points included on the plot (Figure R2). Including more than that would obscure the contours and make the overall trend not apparent.

[Figure]

Figure R2: A scatter plot showing correspondence of Noah layer 1 drying and evaporation rates. Contours are made with all 5+ million points. Displayed in green dots are 1/1000 of the data. 1:1 line is shown in gray. R=0.47.

**Technical Correction (TC) 1:**

Page 4, Line 6-7: Please write out the words first and have the abbreviations in parentheses.

**Technical Response (TR) 1:**

We will make this change.

**TC 2:**

Page 4, Line 13-14: (cm3 cm-3) instead of ", in cm3 cm-3"

**TR 2:**

We will make this change.

**TC 3:**

Page 6 Line 7: "on it" not necessary and sounds a bit awkward.

**TR 3:**

We will omit these words.

**TC 4:**

Page 6, Line 9: The equations appear a bit fuzzy. Is it possible to make these clearer?

**TR 4:**

Yes, we will ensure proper conversion in our revised manuscript.

**TC 5:**

Page 6, Line 16: I believe the cities should be separated with semi-colons, rather than commas (e.g., Fort Cobb, OK; Little River, GA; . . .etc).

**TR 5:**

We will make this change.

**TC 6:**

Page 8, Line 13: covert to convert

**TR 6:**

We will make this change.

---

## Author Response (AR1)

**Final Author Comments**

We thank the two anonymous referees for their helpful and productive comments. We address the comments of the first, followed by the second. For each, we divide comments into general comments and specific comments, and our responses are provided directly after each. After comments and responses, we include the marked-up manuscript (pages 12-39).
* * *
**Referee One:**

**General Comment 1:**

The authors estimate soil drying rates by identifying soil moisture "drydowns" in time series of SMAP observations and Noah land surface model outputs. They compare the estimated drydown rates to covariates such as NDVI, potential evaporation and soil texture, and discuss differences between SMAP and Noah. The study is fairly well written, and the analyses are generally thorough. For example, the authors thoroughly consider important factors such as differences between soil moisture depths of SMAP and Noah in their analyses.

The paper's biggest weakness, in my view, is in its motivation and framing. The story should be tighter. For example, the first line of the abstract says "Drydown periods that follow precipitation events provide an opportunity to assess the mechanisms by which soil moisture dissipates from the land surface." Various mechanisms contribute to soil moisture dissipation – bare soil evaporation, transpiration, percolation, runoff – but only one of these mechanisms is directly estimated from drydowns in the manuscript (bare soil evaporation). It is not clear to me how one would assess the other mechanisms by examining drydowns, and the reader is not provided with any further guidance or results in the manuscript. Perhaps the story could be reframed around estimating bare soil evaporation.

**General Response 1:**

SMAP data can tell us more about bare soil evaporation than other fluxes (transpiration, percolation, and runoff) because (1) the timescale of SMAP observations does not allow for measurements of soil drying until 1 to several days after rainfall -- thus drainage and redistribution have largely ceased (see McColl et al., 2017 for a detailed analysis); and (2) the depth over which the satellite is sensitive to soil moisture necessarily focuses our study on the top few centimeters, where evaporation affects moisture more than transpiration does. Transpiration extracts soil moisture from throughout the entire root zone, usually ~1m thick. These introductory items were presented in the original manuscript on pg 3, lines 3-9, but we will enhance the discussion and be more explicit. We clarify this in the revised manuscript by re-writing much of the introduction (page 2 lines 1-22) and omitting some of the text dedicated to summarizing vadose zone hydrology (originally pg 2, lines 3-13). The linkage to the "loss analysis" in McColl et al. (2017) is strengthened (page 2 lines 9-16). We also add to the abstract (pg 1 lines 12-13), "and our work suggests that SMAP-observed drying is also predominantly affected by direct soil evaporation."

Reference:

McColl, K. A., Wang, W., Peng, B., Akbar, R., Short Gianotti, D. J., Lu, H., Pan, M. and Entekhabi, D.: Global characterization of surface soil moisture drydowns, Geophys. Res. Lett., doi:10.1002/2017GL072819, 2017.

**General Comment 2:**

As far as I can tell, this is essentially a model validation study, except it excludes much of the available data from the analysis (i.e., only includes drydowns). What is gained from doing this that could not be gained from a standard validation study that uses the full time series of data? The authors
10    partially and indirectly answer this question, but the paper would be more compelling if they spent more time directly addressing the paper's motivation. What exactly do we learn from examining soil moisture drydowns in isolation? What specific additional information do we obtain from this analysis compared to a standard validation study that uses all the data, not just drydowns?

**General Response 2:**
15    Our primary goal is to use SMAP data to investigate the processes governing soil drying at a large scale, and then use the resulting information to evaluate some particular aspects of Noah LSM performance. It is certainly important (and feasible) to utilize SMAP to quantify the rate and magnitude of soil wetting during precipitation. We are aware of ongoing studies using soil wetting from SMAP to identify problems with precipitation datasets. This is beyond the scope of our study.
20    Soil drying is a complex process that is of great interest to the hydrologic community (original manuscript pg 1, lines 23-25). The timescale of soil drying affects the fluxes of water, energy, and carbon between the land surface and atmosphere – which is a fundamental aspect of weather and climate documented in many previous studies. Soil drying alone (not wetting) has been the focus of at least three recent studies using SMAP data (Shellito et al., 2016; McColl et al., 2017; McColl et al.,
25    2017b), although the analyses and results described here are very different from the previous papers.
We include Noah LSM in our drying analysis because it allows us to examine the soil drying process in the model. A "standard validation" study would not permit for such a focused analysis of one (critical) aspect of a relatively complex model.
We clarify these objectives in the introduction of the revised manuscript, including (1) the
30    importance of soil drying on land-atmosphere interactions (page 1, lines 26-29); and (2) how this analysis is useful to probe one component of a commonly-used LSM (page 4, lines 6-7).

References:

McColl, K. A., Wang, W., Peng, B., Akbar, R., Short Gianotti, D. J., Lu, H., Pan, M. and
35    Entekhabi, D.: Global characterization of surface soil moisture drydowns, Geophys. Res. Lett., doi:10.1002/2017GL072819, 2017.

McColl, K. A., Alemohammad, S. H., Akbar, R., Konings, A. G., Yueh, S. and Entekhabi, D.: The global distribution and dynamics of surface soil moisture, Nat. Geosci., doi:10.1038/ngeo2868, 2017b.

40    Shellito, P. J., Small, E. E., Colliander, A., Bindlish, R., Cosh, M. H., Berg, A. A., Bosch, D. D., Caldwell, T. G., Goodrich, D. C., McNairn, H., Prueger, J. H., Starks, P. J., van der Velde, R. and

Walker, J. P.: SMAP soil moisture drying more rapid than observed in situ following rainfall events, Geophys. Res. Lett., 43(15), 8068–8075, doi:10.1002/2016GL069946, 2016.

**General Comment 3:**

The lack of a clear story also leads to some strange decisions in the data analysis. In Figure 5, for example, soil moisture drydowns are averaged across space and time, but with only a partial normalization in time, and no normalization in space. Averaging nonlinear (for example, exponential) drydows with varying additive and multiplicative biases would be expected to significantly dampen the nonlinearity and cloud interpretation. Removing the mean of each drydown and scaling VSM to a saturation ratio before averaging would better preserve the expected exponential form of the average drydown. The authors' data analysis choices are not necessarily wrong, but they need to be justified in the context of a larger story that is currently not well-articulated.

**General Response 3:**

The concerns of this reviewer regarding data analysis choices are perhaps caused by a misunderstanding of some of the figures. Specifically, the referee mentions that "In Figure 5, for example, soil moisture drydowns are averaged across space and time…" and expands on how such averaging could cloud interpretation. Figure 5 (and 6 and 9) in the original manuscript are not showing averages, but rather median values from the SMAP data binned according to various criteria. Although we did describe this in the original manuscript (e.g., figure caption for figure 5, text on Page 9, L19 of original manuscript), the review comment shows the point was not made clearly enough. In the revised manuscript, we expand on the method used (page 11, lines 12-13) and remind the reader in the results section that median values were used (page 11, line 18).

The suggestion of "scaling VSM to a saturation ratio," however, is undesirable for our purposes. The calculation of a saturation value would require an estimate of soil texture. Continental-scale soil texture maps are unreliable and this paper shows that Noah's dependence on them is too great (original manuscript, pg 23, line 18).

**General Comment 4:**

I am also concerned about the robustness of the approach used by the authors to estimate drydowns, which relatively frequently produces unphysical, negative evaporative efficiencies. I offer some specific suggestions below.

p. 14, line 12: "SMAP exhibits a considerable number of evaporative efficiency values below zero." Indeed, there is a substantial fraction below zero, and this makes me question the accuracy of the estimates above zero, too. The authors attribute this to noise in the SMAP observations; if it is due to noise, then the authors need to redesign their analyses to be more robust to it. The authors should revisit their drydown algorithm given on p. 7 to ensure this fraction is lower. For instance, they could return to fitting an exponential model, rather than directly estimating soil drying rates from finite differences. They should also alter the criterion "the dry period must be at least 3 days long" (p. 7) to require the dry period to be longer. Three days is quite short given the SMAP revisit time is ~3 days and makes the algorithm highly susceptible to noise in the observations.

**General Response 4:**

SMAP evaporative efficiency values below zero can be attributed to (1) the expected noise in SMAP observations (up to 0.04 cm$^3$ cm$^{-3}$), (2) that soil drying rates trend towards zero in a given dry period (thereby increasing the role of the aforementioned noise), and (3) small amounts of rainfall (that are not large enough to trigger the end of a drydown period or that are not included in the NLDAS forcings) can cause real increases in soil moisture – which appear as 'negative evaporation' when included in the drydown period.

To account for these error sources, we can flag data that exhibit changes of less than 0.04 cm$^3$ cm$^{-3}$, data that constitute the driest 10% of SMAP observations (moisture values below 0.05 cm$^3$ cm$^{-3}$), and data that span even light precipitation. Removing data with those three flags results in a decrease in the number of nonphysical evaporative efficiency values of 98.5%. We revise the manuscript to include this analysis (now page 10, lines 13-16) and move Figure 7 to an earlier position in the manuscript (now Figure 5). To further clarify the purpose of the figure, which is to show that overall, evaporative efficiencies are low, we have changed the figure from 2 panels to 1. Now both SMAP and Noah histograms are shown on the same axes.

While drying values close to or below zero could be excluded from our analysis, it would not change any of the results or conclusions of the analysis. Even if there are noise in the data (within the mission target specified by SMAP), extremely clear relationships exist between soil drying and geophysical variables (original manuscript, Figs 5, 6, and 9). For example, Figure 6a shows clear differences in drying behavior even at values below 0.02 cm$^3$ cm$^{-3}$ day$^{-1}$ (SMAP's noise target for the case of a 2-day overpass). This shows there is value in including SMAP drying increments on the low end of the spectrum. Even with the noise inherent in the ***individual*** observations (i.e., some negative evaporative efficiencies), the large volume of data available provides clear and statistically significant results. Only at the driest soil levels (0.05 cm$^3$ cm$^{-3}$) do the standard errors in Figure 6a (and b and c) extend below 0 cm$^3$ cm$^{-3}$ day$^{-1}$. In the revised manuscript, we clarify the reason for including these negative evaporative efficiencies in page 10, lines 18-20.

Alternative methods of selecting drydowns, such as that used in McColl et al. (2017) or Shellito et al. (2016), include the exact same noise in SMAP observations – the noise just contributes to misfit in the exponential curve fit, rather than negative evaporation values from individual observations. Shellito et al. (2016) found nearly identical results using the exponential fit and finite differences, thus there is no justification for claiming one method is better than the other. In the revised paper (page 10 lines 21-22 and page 11 lines 1-3), we have described the differences between the two approaches, focusing on how noise affects results from each method.

Finally, because each drydown observation is calculated via finite differences, the length of the drydown period is unrelated to the number of "noisy" observations; only two observation go into each calculation, and SMAP overpasses occur every 1 to 3 days. Therefore, drydown periods of only 3 days are still useful and do not disproportionately contribute to noisy observations. We clarify this point in the methods section of the revised manuscript, page 11 lines 4-6.

**Specific Comment (SC) 1:**

p. 3, line 29: "Soil moisture supply, PE rate, and vegetation cover are observed. . ." PE rate is not observed. It is estimated from a reanalysis which is significantly model-Based.

**Specific Response (SR) 1:**

10   We have changed this line to "Soil moisture supply, PE rate, and vegetation cover are observed or calculated …" (page 3, line 30)

**SC 2:**

Fig. 1: The SMAP orbit is clearly visible in Figs 1a and 1b. This should be noted as an unphysical artifact of the observations.

15   **SR 2:**

We added a note to this effect in the caption.

**SC 3:**

p. 7, line 3: how sensitive are the results to choices made in the automated selection process? Comment on this in the manuscript.

20   **SR 3:**

Results are insensitive to minor changes made to the automated selection process. A sentence to this effect has been added to the revised manuscript (pg 7 line 17).

**SC 4:**

p. 8, line 9: "This is an artifact of the SMAP algorithm and does not reflect the drying process." How do
25   the authors know this? Please cite a reference or personal communication, if necessary.

**SR 4:**

We now cite A. Colliander, pers comm. Repeated high soil moisture levels is a feature of the retrieval algorithm when actual moisture levels are above those normally observed (~0.5).

**SC 5:**

30   p. 8, line 21: "Noah drying rates have a nearly 1:1 relationship with evaporation rates." The relationship is substantially noisy with $R = 0.47$, and clearly not "1:1". Perhaps rephrase to something like "the slope of the regression line is nearly 1".

**SR 5:**

We have changed this wording in the revised manuscript (page 9 line 9) and added a best-fit line to the
35   figure.

**SC 6:**

[Specific comment 6 was copied into General Comment 4 and responded to in General Response 4].

**SC 7:**

p. 15, line 8: "vegetation amount (as indicated by NDVI)" NDVI is a measure of greenness, not vegetation amount. This is only loosely correlated with vegetation amount. Please add nuance to this statement. Also, since NDVI is an optical index and is therefore not observed during cloudy conditions, how do the authors expect this to impact their results? For example, cloudy conditions will lower PE, all else being equal; therefore these conditions are being systematically excluded from the comparison. Comment on this in the manuscript.

**SR 7:**

The NDVI product we use is a 16-day product. Each individual day's value is an interpolation between non-cloudy conditions. (See original manuscript pg 5, lines 18-20.) Therefore, cloudy conditions do not affect the results of this study.

**SC 8:**

p. 19, line 6: ". . .are completely independent from the Noah model results." NDVI and FG are hardly independent of one another: FG is estimated directly from NDVI!

**SR 8:**

Thank you for this comment. This section has been re-written. Our intent is to point out that SMAP results come from observations and an algorithm that does not incorporate Noah simulations. In contrast, Noah simulations ingest Fg, which (as the referee points out) have been estimated directly from NDVI.

Alternative text has been inserted into the revised manuscript, page 16, lines 6-10:

"SMAP drying dynamics are dependent on environmental states observed at the time of each overpass. Noah, however, depends on a vegetation parameter that has been calculated from climatology: its $F_G$ parameter is derived from 5 years of NDVI observations (Gutman and Ignatov, 1998). In addition, Noah results reflect the behavior of the 10 cm surface layer, whereas SMAP nominally senses only the top 5 cm. These two inconsistencies must be considered carefully to fairly compare SMAP and Noah results."

* * *
**Referee Two:**

**General Comment:**

This article calculates drying rates over parts of North America and assesses the relative roles of other land surface characteristics such as vegetation and soil texture in soil dry down. This paper extends previous work by expressing soil dry down in multiple ways and by comparing to Noah land surface model simulations. A key finding is that SMAP dries down more quickly after precipitation than Noah and that evaporative efficiency is reduced when vegetation is increased.

The article is well-written and will certainly be of interest to the land community. The methods and the results presented here are useful from a soil science perspective but also seem likely to be helpful in better understanding and improving land surface models. I offer minor comments below to improve the readability of the manuscript.

**Specific Comment (SC) 1:**

Page 1, Line 10-11: "Data cover the domain of the NLDAS 2". Please reword – just because the data cover this domain, doesn't meant that the whole domain will be used (as is the case here).

**Specific Response (SR) 1:**

The domain used in this study is in fact the NDLAS-2 domain.

**SC 2:**

Page 3 Line 6 – Although the nominal SMAP depth is 5 cm, the sensing depth also changes slightly with moisture content. Is it possible that this could affect the conclusions?

**SR 2:**

Yes it is possible that the varying sensing depth could affect the conclusions. We have added a sentence to this effect in the revised manuscript, page 3 lines 12-13: "...will propagate down to greater depths. In addition, L-band sensing depth varies slightly with moisture content (Njoku and Kong 1977), though quantifying the effect of this change is beyond the scope of this study." In addition, we add the phrase "and reach a nominal sensing depth of 5 cm" to pg 4, line 19 to clarify that this is not an exact sensing depth. In the discussion, we add the following to the manuscript (page 22, lines 23-26): "In addition, it is possible that SMAP drying rates are exaggerated due to slight decreases in L-band sensing depth that accompany wet soil (Njoku and Kong 1977). After rainfall, moisture in the top couple centimeters could dominate the signal, leading to the entire 0-5 cm sensing depth being assigned a moisture level that is only present at the very surface. As the soil dries and becomes more evenly distributed within the sensing depth, such abnormalities would dissipate."

**SC 3:**

Page 6 Line 24: Was any information lost in the re-gridding? How do you know (comparison of statistics to the original, etc)?

**SR 3:**

The analysis in this study would not have been possible without some sort of assignation of MODIS and NLDAS-2 pixels into the SMAP EASE-2 grid (regridding). The native grid for NDVI is finer than the SMAP grid, so there is concern for loss of information in that conversion. To that end, we compare empirical PDFs of average annual NDVI before and after re-gridding. The native and re-gridded data are nearly indistinguishable from one another (Figure R1).

The native NLDAS-2 grid is slightly coarser than the SMAP grid. This means that the conversion did not lose any data, but rather that information was occasionally repeated, as described in the original manuscript pg 6, lines 27-30: "The NLDAS-2 grid is only slightly coarser than SMAP's grid, so occasionally the same data will be mapped into two SMAP pixels. Though this is not ideal, it is preferable to basing our analysis on the NLDAS-2 grid, which would force us to exclude some SMAP pixels or blend them with their neighbor when they fall within the same NLDAS-2 pixel."

[Figure]

Figure R1: Empirical probability distribution functions of native and re-gridded NDVI data.

**SC 4:**

Page 10, Section 2.2.5: This section is very short and doesn't seem substantial enough to be its own section. Perhaps move the information to 2.1.5.

**SR 4:**

Thank you for this suggestion. We have moved the information to 2.1.5 to increase readability.

**SC 5:**

Page 14 line 14: I don't see the wetting between successive overpasses in Figure 2. Is it possible to point out a time period as an example?

**SR 5:**

We have added, "... as can be seen for several cases in the upper panels of Figure 2: late November at Fort Cobb, OK, and mid October at Marena, OK."

**SC 6:**

Figure 4: Is it possible to also include the points on this plot, rather than just the contours?

**SR 6:**

We have recreated Figure 4 with 1 in 1000 points included on the plot (Figure R2). Including more than that would obscure the contours and make the overall trend not apparent.

[Figure]

Figure R2: Scatter and contour plot showing correspondence of Noah layer 1 drying and evaporation rates. Displayed green markers show 0.1 % of the 5+ million data points. Contours are drawn using all data. 1:1 line is shown in solid gray. Best fit line shown in dashed gray. R=0.47.

**Technical Correction (TC) 1:**

Page 4, Line 6-7: Please write out the words first and have the abbreviations in parentheses.

**Technical Response (TR) 1:**

We have made this change.

**TC 2:**

Page 4, Line 13-14: (cm3 cm-3) instead of ", in cm3 cm-3"

**TR 2:**

We have made this change.

**TC 3:**

Page 6 Line 7: "on it" not necessary and sounds a bit awkward.

**TR 3:**

We have omitted these words.

**TC 4:**

Page 6, Line 9: The equations appear a bit fuzzy. Is it possible to make these clearer?

**TR 4:**

Yes, we ensure proper conversion in our revised manuscript.

**TC 5:**

Page 6, Line 16: I believe the cities should be separated with semi-colons, rather than commas (e.g., Fort Cobb, OK; Little River, GA; . . .etc).

**TR 5:**

We have made this change.

**TC 6:**

Page 8, Line 13: covert to convert

**TR 6:**

We have made this change.

[revised manuscript text omitted]

---

## Author Response (AR2)

**Author Comments**

The editor has requested the following changes. Author responses are provided directly after each.

(1) Please provide a response to Reviewer #1's comment regarding Figure 5.

Referee #1's comment regarding Figure 5 is copied in full here:

"The lack of a clear story also leads to some strange decisions in the data analysis. In Figure 5, for example, soil moisture drydowns are averaged across space and time, but with only a partial normalization in time, and no normalization in space. Averaging nonlinear (for example, exponential) drydowns with varying additive and multiplicative biases would be expected to significantly dampen the nonlinearity and cloud interpretation. Removing the mean of each drydown and scaling

VSM to a saturation ratio before averaging would better preserve the expected exponential form of the average drydown. The authors' data analysis choices are not necessarily wrong, but they need to be justified in the context of a larger story that is currently not well-articulated."

We addressed these concerns previously in General Response 3. We expand on that response here.

First, the figure shows medians, not averages. We have clarified this fact in both the methods (page 11, line 13) and results (page 11, line 18).

Second, exponential decay is indeed a useful model for characterizing the drying of soil moisture, as described in the introduction of this manuscript (page 3, lines 18-21) and in (McColl et al., 2017; Shellito et al., 2016). However, by calculating linear drying rates between pairs of sequential observations, we avoid introducing new uncertainties into the analysis associated with imposing an exponential decay model onto the system. This allows us to analyze soil drying using dynamic controls that can vary within a single drydown period instead of static controls that must reflect conditions over the entirely of the drydown event or location (page 3, lines 28-33).

Our analysis simplifies the drying process into a collection of linear calculations. Importantly, this methodology does not produce "dampened" behavior in the aggregate. Rather, Figure 6 shows that our aggregate analysis preserves a drydown shape reminiscent of exponential decay and therefore supports the introductory statement that the two analysis methods offer similar information via different means (page 3, lines 26-28).

Lastly, the suggestion of "scaling VSM to a saturation ratio," is undesirable, since the calculation of a saturation value will introduce additional uncertainties into our analysis.

i

Thank you. We have removed "very."

[revised manuscript text omitted]